# Node Importance Specific Meta Learning in Graph Neural Networks

## Abstract

While current node classification methods for graphs have enabled significant progress in many applications, they rely on abundant labeled nodes for training. In many real-world datasets, nodes for some classes are always scarce, thus current algorithms are ill-equipped to handle these few-shot node classes. Some meta learning approaches for graphs have demonstrated advantages in tackling such few-shot problems, but they disregard the impact of node importance on a task. Being exclusive to graph data, the dependencies between nodes convey vital information for determining the importance of nodes in contrast to node features only, which poses unique challenges here. In this paper, we investigate the effect of node importance in node classification meta learning tasks. We first theoretically analyze the influence of distinguishing node importance on the lower bound of the model accuracy. Then, based on the theoretical conclusion, we propose a novel Node Importance Meta Learning architecture (NIML) that learns and applies the importance score of each node for meta learning. Specifically, after constructing an attention vector based on the interaction between a node and its neighbors, we train an importance predictor in a supervised manner to capture the distance between node embedding and the expectation of same-class embedding. Extensive experiments on public datasets demonstrate the state-of-the-art performance of NIML on few-shot node classification problems.

## 1 Introduction

Graph structure can model various complicated relationships and systems, such as molecular structure (Subramanian et al., 2005), citationships (Tang et al., 2008b) and social media relationships (Ding et al., 2019). The use of various deep learning methods (Hamilton et al., 2017; Kipf & Welling, 2016) to analyze graph structure data has sparked lots of research interest recently, where node classification is one of the essential problems. Several types of graph neural networks (GNNs) (Veličković et al., 2017; Wu et al., 2020) have been proposed to address the problem by learning high-level feature representations of nodes and addressing the classification task end-to-end.

Despite the success in various domains, the performance of GNNs drops dramatically under the few-shot scenario (Mandal et al., 2022), where extremely few labeled nodes are available for novel classes. For example, annotating nodes in graph-structured data is challenging when the samples originate from specialist disciplines (Guo et al., 2021) like biology and medicine.

Many meta learning works, including optimization-based methods (Finn et al., 2017) and metric-based methods (Snell et al., 2017; Vinyals et al., 2016), have demonstrated their power to address few-shot problems in diverse applications, such as computer vision and natural language processing (Lee et al., 2022). In meta learning, a meta learner is trained on various tasks with limited labeled data in order to be capable of fast generalization and adaption to a new task that has never been encountered before. However, it is considerably challenging to generalize these meta learning algorithms designed for independent and identically distributed (i.i.d.) Euclidean data to graph data.

To address the few-shot node classification problem, some graph meta learning approaches have been proposed (Liu et al., 2021; Ding et al., 2020; Yao et al., 2020). They structure the node classification problem as a collection of tasks. The key idea is to learn the class of nodes in the query set by transferring previous knowledge from limited support nodes in each task. However, most

existing approaches simply assume that all labeled nodes are of equal importance to represent the class they belong to. Differences and interdependencies between nodes are not considered in the learning process of the few-shot models. Since only limited data points are sampled to generate tasks in meta learning, each sampled task has high variance; therefore, treating all the data points equally might lead to loss of the crucial information supplied by central data points and render the model vulnerable to noises or outliers. In particular, the relationship between nodes and neighbors in a graph is an important factor that carries node information in addition to node features, and can be utilized as a starting point to investigate the importance of nodes. Although some work (Ding et al., 2020) considers the importance of nodes, there is lack of theoretical analysis about it.

To address the aforementioned challenges, we first explore, in a theoretical manner, the effect of distinguishing nodes of different degree of importance on the lower bound of the accuracy of the model. We analyze the ProtoNet (Snell et al., 2017), and conclude that when important nodes are given more weight when computing prototype representations in a task, the prototype will get closer to its own expectation, thus the lower bound of the accuracy will be increased. Based on this theoretical result, we propose a node importance meta learning framework (NIML) for learning and using the node importance in a task. Specifically, an attention vector is constructed for each node to describe the relationship distribution of that node and its neighbors. Then we train a supervised model using this attention vector as input to learn the distance between the node embedding and the same-class prototype expectation, effectively capturing the importance of that node to its class. The obtained distance will be used to calculate a weighted prototype in meta learning. We conduct experiments on three benchmarks, and results validate the superiority of proposed NIML framework.

To summarize, the main contributions of this paper are as follows: 1) We theoretically explore the influence of node importance on the lower bound of model accuracy and show the benefit of distinguishing between nodes of different importance in a meta learning task. The theory conclusion can be applied to any domain, not only graph data. 2) We design a category-irrelevant predictor to estimate the distance between node embedding and approximated prototype expectation and follow the theorem conclusion to compute a weighted prototype, where we construct an attention vector as the input, which describes the distribution of neighbor relationships for a given node. 3) We perform extensive experiments on various real-world datasets and show the effectiveness of our approach.

## 2 RELATED WORKS

### 2.1 GRAPH NEURAL NETWORKS

Recent efforts to develop deep neural networks for graph-structured data have been largely driven by the phenomenal success of deep learning (Cao et al., 2016; Chang et al., 2015). A large number of graph convolutional networks (GCNs) have been proposed based on the graph spectral theory. Spectral CNN (Bruna et al., 2013) mimics the properties of CNN by defining graph convolution kernels at each layer to form a GCN. Based on this work, researches on GCNs are increasingly getting success in (Defferrard et al., 2016; Henaff et al., 2015; Kipf & Welling, 2016). Graph Attention Networks (GATs) (Veličković et al., 2017) learn the weights of node neighbors in the aggregation process by an attention mechanism. GraphSAGE (Hamilton et al., 2017) utilizes aggregation schemes to aggregate feature information from local neighborhoods. However, modern GNN models are primarily concerned with semi-supervised node classification. As a result, we develop a GNN framework to address the few-shot difficulties in graph data, which is one of their largest obstacles.

### 2.2 META LEARNING

Existing meta learning algorithms mainly fall into two categories (Hospedales et al., 2020): optimization-based meta learning and metric-based meta learning. Optimization-based meta learning (Finn et al., 2017; Li et al., 2017; Mishra et al., 2017; Ravi & Larochelle, 2016; Mishra et al., 2017) aims to learn an initialization of parameters in a gradient-based network. MAML (Finn et al., 2017) discovers the parameter initialization that is suitable for various few-shot tasks and can be used in any gradient descent model. MetaSGD (Li et al., 2017) advances MAML and learns the initialization of weights, gradient update direction, and learning rate in a single step. Metric-based meta learning (Liu et al., 2019; Ren et al., 2018; Snell et al., 2017; Sung et al., 2018; Vinyals et al., 2016) focuses on learning a generalized metric and matching function from training tasks. In partic-

ular, Prototypical Networks (ProtoNet) (Snell et al., 2017) embed each input into a continuous latent space and carry out classification using the similarity of an example to the representation of latent classes. Matching Networks (Vinyals et al., 2016) learn a weighted nearest-neighbor classifier with attention networks. Ren et al. (2018) propose a novel extension of ProtoNet that are augmented with the ability to use unlabeled examples when producing prototypes. Relation Network (Sung et al., 2018) classifies new classes by computing a relation score between the query set and a few samples in each new class. Most existing meta learning methods cannot be directly applied to graph data due to lack of the ability to handle node dependencies.

### 2.3 FEW SHOT LEARNING ON GRAPHS

Current node representation learning cannot handle unseen classes with few-shot data. Some few-shot research on graphs target on node/link/graph classification (Mandal et al., 2022). We introduce the node classification works as follows. Meta-GNN (Zhou et al., 2019) extends MAML (Finn et al., 2017) to graph data. RALE (Liu et al., 2021) considers the dependency between nodes within a task and alignment between tasks, then learns the hub-based relative and absolute location embedding. G-Meta (Huang & Zitnik, 2020) uses a local subgraph to represent the nodes given local structural information. MetaHG (Qian et al., 2021) presents a heterogeneous graph few-shot learning model for automatically detecting illicit drug traffickers on Instagram. MetaTNE (Lan et al., 2020) combines the skip-gram mechanism with meta learning to capture the structural information with known labels and without node attributes. GFL (Yao et al., 2020) implements few-shot classification on unseen graphs for the same set of node classes. GPN (Ding et al., 2020) aggregates node importance scores and learns node embedding with a few-shot attributed network based on ProtoNet. However, a theoretical analysis of the effect of node importance on meta learning is still missing.

## 3 PRELIMINARY

### 3.1 META LEARNING PROBLEM SETUP

We first introduce some notations of few-shot classification problems. Let $C$ be the space of classes with a probability distribution $\tau$, and $\chi$ be the space of input data. We sample $N$ classes $c_1, \cdots, c_N$ i.i.d form $\tau$ to form an $N$-way classification problem. For each class $c_i$, $k$ data points are sampled as $\mathbb{S}_i = \{_s\boldsymbol{x}_1, \cdots, _s\boldsymbol{x}_k | (_s\boldsymbol{x}_j, _sy_j) \in \chi \times C \cap (_sy_j = c_i)\}$ to constitute the support set, where $_s\boldsymbol{x}_j \in \mathbb{R}^D$, $D$ is the dimension of input data, $_sy_j$ is the class of $_s\boldsymbol{x}_j$. Thus the support set is a union of $\mathbb{S}_i$, and $\mathbb{S} = \cup_{i=1}^N \mathbb{S}_i$. Besides, for each class $c_i$, we sample $m$ data points to form a part of query set $\mathbb{Q}$ in the same way. The table of notation and definition can be found in the appendix.

The core idea of meta learning algorithms is to train on various tasks sampled from distribution $\tau$ and then equip the model with the ability to fast generalize and adapt to unseen tasks with limited labeled data. Each $N$-way $k$-shot task is sampled by the above method. In the meta-train phase, ground truth of $\mathbb{S}$ and $\mathbb{Q}$ are both known, and $\mathbb{Q}$ is used to evaluate the performance of model updated by $\mathbb{S}$. During the meta-test phase, the performance of the model will be evaluated on unseen classes. We assume each unseen class follows the same distribution $\tau$.

### 3.2 PROTOTYPICAL NETWORKS

ProtoNet (Snell et al., 2017) is a metric-based meta learning algorithm. It learns an embedding function $f_\phi : \mathbb{R}^D \to \mathbb{R}^M$, which maps input data from $\chi$ to the embedding space. The $M$-dimensional prototype representation $\overline{\boldsymbol{c}_i}$ for each class $c_i$ is computed by averaging the embedding of all data points belonging to $c_i$ in the support set:

$$\overline{\boldsymbol{c}_i} = \frac{1}{|\mathbb{S}_i|} \sum_{j=1}^k f_\phi(_s\boldsymbol{x}_j). \tag{1}$$

Given a distance function $d(\boldsymbol{x}, \boldsymbol{x}')$, the probability a data point $\boldsymbol{x}$ belongs to class $n$ is calculated by Softmax function over squared distance between the embedding of $\boldsymbol{x}$ and prototype representations.

$$p_\phi(y = n | \boldsymbol{x}) = \frac{exp(-d(f_\phi(\boldsymbol{x}), \overline{\boldsymbol{c}_n}))}{\sum_{j=1}^N exp(-d(f_\phi(\boldsymbol{x}), \overline{\boldsymbol{c}_j}))}. \tag{2}$$

The prediction of an input $\boldsymbol{x}$ is computed by taking argmax over probability function $p_\phi(y = n|\boldsymbol{x})$. Let $\hat{y}$ be the prediction of an input $\boldsymbol{x}$, then $\hat{y} = \arg\max_j(p_\phi(y = j|\boldsymbol{x}))$. The loss function for input data belongs to class $n$ is in the form of negative log-likelihood $J(\phi) = -log(p_\phi(y = n|\boldsymbol{x}))$. Thus, the parameters of embedding function $f_\phi$ is updated by minimizing the sum of loss functions on query sets. After the process of meta learning, the function $f_\phi$ has the ability to embed data points belonging to the same class to the same group in the embedding space $\mathbb{R}^M$.

## 4 THEORETICAL ANALYSIS

In this section, we use ProtoNet (Snell et al., 2017), a classic metric-based meta learning algorithm as an example, to theoretically explore the effect of node importance on the lower bound of model accuracy in the embedding space. The **theoretical conclusion** is that assigning higher weight to the data point that has closer distance to the prototype expectation will increase the lower bound of accuracy. This conclusion thus motivates us to use abundant data to learn the distance between node representation and prototype expectation in NIML framework.

We derive our theorem based on a previous work (Cao et al., 2019). The detailed proof process is included in the Appendix A.1. We first define the expected accuracy $R$ of $\phi$ as:

$$R(\phi) = \mathbb{E}_c \mathbb{E}_{\mathbb{S}, \boldsymbol{x}, y} I\left[\arg\max_j \{p_\phi(\hat{y} = j \mid \boldsymbol{x}, \mathbb{S})\} = y\right],\tag{3}$$

where $I$ denotes the indicator function.

In order to simplify the theorem, we present the analysis for a special case: 2-way 2-shot problem i.e. a binary classification with 2 nodes for each class. Note that the theorem we present can also be extended to an $N$-way $k$-shot problem. We adopt the assumption that for any input $\boldsymbol{x}$ in each class $c$, the embedding vector $f_\phi(\boldsymbol{x})$ follows a Gaussian distribution, where $p(f_\phi(\boldsymbol{x}) \mid y = c) = \mathcal{N}(\mu_c, \Sigma_c)$. $\mu_c$ is the expectation of $f_\phi(\boldsymbol{x})$ when the input $\boldsymbol{x}$ belongs to class $c$, and $\Sigma_c$ is the expected intra-class variance of class $c$. We denote $\Sigma$ as the variance between classes.

**Define importance based on prototype deviation**: We want to explore the influence of differentiating data with different degrees of importance on the accuracy $R$. Since only a few data points are sampled for one class to form a task, when we compute $\overline{c_i}$ following Equation( 1), there exists deviation between $\overline{c_i}$ and $\mu_i$. As we simplify the problem to a 2-shot setting, the embedding vector of two nodes belonging to the class $c_i$ can be denoted by $\mu_i - \epsilon_1$ and $\mu_i + \epsilon_2$ respectively. We would like to emphasize that the sign of $\epsilon_i$ can be permuted freely and will have no effect on the theorem. After that, we naturally treat the node which has an embedding vector that is closer to the expectation $\mu_i$ as the more important node. Based on this consideration, we redefined the prototype calculation as below.

**Definition 1** *We change the definition of $\overline{c_i}$ to a weighted form. Let $\boldsymbol{x}_1$ and $\boldsymbol{x}_2$ be the feature vector of two nodes belonging to class $c_i$. The embedding of $\boldsymbol{x}_1$ and $\boldsymbol{x}_2$ is: $f_\phi(\boldsymbol{x}_1) = \mu_i - \epsilon_1$, and $f_\phi(\boldsymbol{x}_2) = \mu_i + \epsilon_2$. $w_1$ and $w_2$ are weights related to $f_\phi(\boldsymbol{x}_1)$ and $f_\phi(\boldsymbol{x}_2)$, which can be either trainable or pre-defined. Then,*

$$\overline{c_i} = \frac{w_1}{w_1 + w_2}f_\phi(\boldsymbol{x}_1) + \frac{w_2}{w_1 + w_2}f_\phi(\boldsymbol{x}_2).\tag{4}$$

*When $w_1 = w_2$ in Equation( 4), Equation( 4) is equivalent to Equation( 1).*

We would like to prove our **key idea**: in Definition 1, when $w_1$, $w_2$ and $\epsilon_1$, $\epsilon_2$ have opposite relative value relationships (i.e. If $w_1 > w_2$, $\epsilon_1 < \epsilon_2$), which means greater weight is assigned to the more important node, this setting allows the lower bound of the model to be raised. Some theoretical results are provided below, and the whole proof is included in the Appendix.

Let $a$ and $b$ denote the two classes sampled from $\tau$ for a task. Since all classes follow the same distribution, we only need to select one class and investigate the model accuracy for each node inside this class and extend the results to remaining classes. Let $\boldsymbol{x}$ be the feature of a node drawn from class $a$, then Equation( 3) can be written as:

$$R(\phi) = \mathbb{E}_{a,b \sim \tau} \mathbb{E}_{\mathbf{x} \sim a, \mathbb{S}} I[\hat{y} = a].\tag{5}$$

**Proposition 1** *We can express Equation( 5) as a probability function:*

$$R(\phi) = \Pr{}_{a,b,\boldsymbol{x},\mathbb{S}}(\hat{y} = a) = \Pr{}_{a,b,\boldsymbol{x},\mathbb{S}}(\alpha > 0), \tag{6}$$

*where $\alpha \triangleq \|f_\phi(\boldsymbol{x}) - \overline{\boldsymbol{c}_b}\|^2 - \|f_\phi(\boldsymbol{x}) - \overline{\boldsymbol{c}_a}\|^2$.*

*From the one-sided Chebyshev's inequality, it can be derived that:*

$$R(\phi) = \Pr(\alpha > 0) \geqslant \frac{\mathbb{E}[\alpha]^2}{\mathrm{Var}(\alpha) + \mathbb{E}[\alpha]^2}. \tag{7}$$

**Lemma 1** *Consider space of classes $C$ with sampling distribution $\tau$, $a, b \overset{iid}{\sim} \tau$. Let $\mathbb{S} = \{\mathbb{S}_a, \mathbb{S}_b\}$ $\mathbb{S}_a = \{{}_a\boldsymbol{x}_1, \ldots, {}_a\boldsymbol{x}_k\}, \mathbb{S}_b = \{{}_b\boldsymbol{x}_1, \ldots, {}_b\boldsymbol{x}_k\}, k \in \mathbb{N}$ is the shot number, and $y(\boldsymbol{x}) = a$. Define $\overline{\boldsymbol{c}_a}$ and $\overline{\boldsymbol{c}_b}$ as shown in Equation( 4). Then,*

$$\mathbb{E}_{\boldsymbol{x},\mathbb{S}|a,b}[\alpha] = (\mu_a - \mu_b)^T(\mu_a - \mu_b) + (2\mu_b + \sigma_b - 2\mu_a)^T\sigma_b + \sigma_b^T\sigma_b - \sigma_a^T\sigma_a, \tag{8}$$

$$\mathbb{E}_{a,b,\boldsymbol{x},\mathbb{S}}[\alpha] = 2\,\mathrm{Tr}(\Sigma) + \sigma_b^T\sigma_b - \sigma_a^T\sigma_a, \tag{9}$$

$$\mathbb{E}_{a,b}[\mathrm{Var}(\alpha \mid a, b)] \leq 8\left(1 + \frac{1}{k}\right)\mathrm{Tr}\{\Sigma_c\left(\left(1 + \frac{1}{k}\right)\Sigma_c + 2\Sigma\right) + \sigma_b^T\sigma_b + \sigma_a^T\sigma_a\}, \tag{10}$$

*where*

$$\sigma_a = \frac{{}_aw_{2a}\epsilon_2 - {}_aw_{1a}\epsilon_1}{{}_aw_2 + {}_aw_1}, \sigma_b = \frac{{}_bw_{2b}\epsilon_2 - {}_bw_{1b}\epsilon_1}{{}_bw_2 + {}_bw_1}$$

Lemma 1 provides several key components for Theorem 1. Two new variables are introduced: $\sigma_a$ and $\sigma_b$, defined by $\sigma_a = \overline{\boldsymbol{c}_a} - \mu_a$ and $\sigma_b = \overline{\boldsymbol{c}_b} - \mu_b$.

**Theorem 1** *Under the condition where Lemma 1 hold, we have:*

$$R(\phi) \geqslant \frac{(2\,\mathrm{Tr}(\Sigma) + \sigma_b^T\sigma_b - \sigma_a^T\sigma_a)^2}{f_1(\sigma_a, \sigma_b) + f_2(\sigma_a, \sigma_b)}, \tag{11}$$

*where*

$$f_1(\sigma_a, \sigma_b) = 12\,\mathrm{Tr}\{\Sigma_c(\frac{3}{2}\Sigma_c + 2\Sigma + \sigma_b^T\sigma_b + \sigma_a^T\sigma_a)\}$$

$$f_2(\sigma_a, \sigma_b) = \mathbb{E}_{a,b}[((\mu_a - \mu_b)^T(\mu_a - \mu_b) + (2\mu_b + \sigma_b)^T\sigma_b)^2].$$

The lower bound of model accuracy $R(\phi)$ is in the form of a fraction, where we denote the denominator using the sum of two functions $f_1(\sigma_a, \sigma_b)$ and $f_2(\sigma_a, \sigma_b)$. We would like to investigate the effect of a change in $\sigma_a, \sigma_b$ on $R(\phi)$, where $\sigma_a, \sigma_b$ are the bias between $\mu_a, \mu_b$ and $\overline{\boldsymbol{c}_a}, \overline{\boldsymbol{c}_b}$. From the definition in Lemma 1, we can divide $\sigma_c$ for a class $c$ into three cases: If $w$ and $\epsilon$ are negatively correlated, the value of $\sigma_c$ is closest to 0 among the three cases; If the same $w$ is given to each $\epsilon$, this corresponds to the case of calculating the prototype directly with the average embedding value. If $w$ and $\epsilon$ are positively correlated, which is an opposite case from the first one, the value of $\sigma_c$ is farthest from 0. We emphasize that for all classes in one episode, they have the same assignment strategy, thus $\sigma_a$ and $\sigma_b$ are positively correlated.

According to Theorem 1, we notice that $\sigma_a$ and $\sigma_b$ always appear in the form of a squared norm; thus, their positives or negatives have little effect on the result. In the numerator, $\sigma_b^T\sigma_b$ and $\sigma_a^T\sigma_a$ are subtractive, whereas they are additive in the denominator. After analyzing their degree and coefficients, we can reach the following **conclusion**: when we use the first strategy to assign values for $w$ and $\epsilon$, the lower bound of accuracy $R(\phi)$ will be improved. In detail, when $w$ and $\epsilon$ are negatively correlated, $\sigma_a$ and $\sigma_b$ are both closest to 0, resulting in an increase in the value of lower bound. This theoretical result is exactly in line with our perception: when the value of $\sigma_a$ and $\sigma_b$ are close to 0, it means that the prototype embedding we compute with the weighted node embedding is very close to its expectation $\mu_a$ and $\mu_b$, which is what we anticipate the prototype should achieve. Besides, from $f_2(\sigma_a, \sigma_b)$, we can conclude that bringing $\sigma_b$ close to 0 will help reduce the sensitivity of the lower bound to $\mu_b$. Thus, if the distance $\epsilon$ between given data point and prototype expectation could be predicted, the weight can be assigned by the first strategy to enhance the model accuracy.

## 5 FRAMEWORK

Inspired by theoretical results, we propose to prioritize node importance in graph meta learning problems by introducing an importance score predictor. In detail, by constructing an attention vector to describe the relationship distribution of a given node, we end-to-end predict the distance between node embedding and prototype expectation, which is further used to compute a weighted average of node embeddings as the more accurate prototype representation.

### 5.1 FEW-SHOT NODE CLASSIFICATION TASK

We denote an undirected graph as $\mathcal{G} = (V, E, \boldsymbol{A}, \boldsymbol{X})$, where $V = \{v_1, \cdots, v_n\}$ is the node set, $E = \{e_1, \cdots, e_m\}$ is the edge set. The adjacency matrix $\boldsymbol{A} = \{0, 1\}^{n \times n}$ represents the graph structure, where $a_{ij}$ denotes the weight between node $v_i$ and $v_j$. $\boldsymbol{X} \in \mathbb{R}^{n \times d}$ is the feature matrix, where $\boldsymbol{x}_i \in \mathbb{R}^d$ represents the feature of node $v_i$.

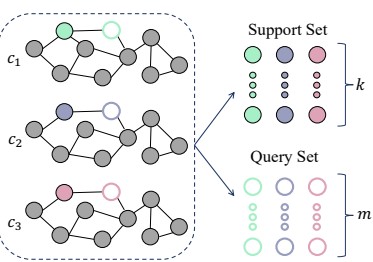

We focus on solving **few-shot node classification** problems. Episode training is adopted in the meta-train phase as previous works Snell et al. (2017), which samples several tasks and updates parameters based on the sum of the loss functions of the query sets. In our problem, nodes in the graphs correspond to data points in Euclidean space, and an $N$-way $k$-shot problem implies that each of the $N$ categories has $k$ nodes. The query set and support set are illustrated in Figure 1.

Figure 1: An episode for a 3-way $k$-shot problem. $c_i$ represents node class, $k$ and $m$ denote the number of nodes for each class in support set $\mathbb{S}$ and query set $\mathbb{Q}$.

### 5.2 NODE REPRESENTATION LEARNING

Our graph prototypical network has a node representation learning component. Following the idea from ProtoNet (Snell et al., 2017) introduced in Section 3, we aim to train an embedding function $f_\theta(v_i, \boldsymbol{x}_i)$ that learns the node representation of $v_i$, thus prototypes representing each category of the task can be computed. The node classification can then be implemented by calculating the distance between the current node and each prototype.

On graph data, the embedding function is implemented with an inductive Graph Neural Network (GNN) (Hamilton et al., 2017) that learns a low-dimensional latent representation of each node. It follows a neighborhood combination and aggregation scheme, where each node recursively fetches information from its neighbors layer by layer. Let $\boldsymbol{h}_v^l$ denote a node $v$'s representation at the $l^{\text{th}}$ step,

$$\begin{aligned} \boldsymbol{h}_{N(v)}^l &= \text{AGGREGATE}_l(\boldsymbol{h}_u^{l-1}, \forall u \in N(v)), \\ \boldsymbol{h}_v^l &= \sigma(\boldsymbol{W}^l \cdot \text{CONCAT}(\boldsymbol{h}_v^{l-1}, \boldsymbol{h}_{N(v)}^l)), \end{aligned} \tag{12}$$

where $N(v)$ represents node $v$'s (sampled) neighbors. The first step is to aggregate the representations of neighbor nodes in layer $l - 1$ into a new vector $\boldsymbol{h}_{N(v)}^l$. The node representation on layer $l - 1$ and the aggregated neighborhood representation are concatenated, which is then fed to a fully connected layer with nonlinear activation function $\sigma$. We denote this $L$-layer GNN by $f_\theta(\cdot)$.

### 5.3 NIML: NODE IMPORTANCE SPECIFIC PROTOTYPICAL NETWORK

Prototype is typically calculated by averaging node embeddings inside the support set as Equation( 1) shows. However, based on our theoretical findings, distinguishing nodes of different importance within a category can increase the model accuracy. When the number of nodes in the task is relatively small, the deviation produced by randomly sampling nodes for the prototype computation can be reduced by assigning higher weights to nodes with more importance (i.e. less distance to the prototype expectation). We therefore develop a model to learn the importance score of each node, which contributes to a **weighted** prototype computation.

Although the theory motivates us to assign weights according to the distance between the node representation and the prototype expectation, it is based on the assumption that the distance $\epsilon$ is known. To overcome this obstacle, we design a model which end-to-end **predicts the distance**.

Since numerous tasks are sampled during meta-train phase, we get access to relatively abundant nodes belonging to each class. When the number of nodes in a category is large enough, the prototype expectation $\mu_c$ can be approximated by the mean embedding of same-class nodes among the whole graph, where $\mu_c \simeq mean(f_\phi(\boldsymbol{x}_u))$, for each node $u$ belongs to class $c$. Then the ground truth distance $\epsilon$ between a node $v$ and its same-class prototype expectation can be computed by $d_{vp} = d(f_\phi(\boldsymbol{x}_v), \mu_c)$. Thus, theoretically speaking, we expect that the distance function can be learned with the iterative meta-training.

The next step is to decide which node information should be used to predict the distance. Directly using node embedding generated by Proto-GCN as input does not meet our expectation for distance predictor. Proto-GCN maps same-class nodes to close locations in the embedding space; whereas distance predictor maps nodes of comparable importance to close distance value, so nodes of different categories may be mapped to the same location (as shown in Figure 6 in Appendix A.3). Hence, it is necessary to design an input which containing as little label information as possible.

Due to the feature smoothing mechanism of GNN, an $L$-layer GNN brings the same smooth intensity for each node. Assuming we consider the homophily graph, the neighboring nodes have similar features. With equal smooth intensity, the similarity between a central node and its neighbors is higher than that between a marginal node and its neighbors, thus the relationship between a central node and its neighbors is more uniformly distributed.

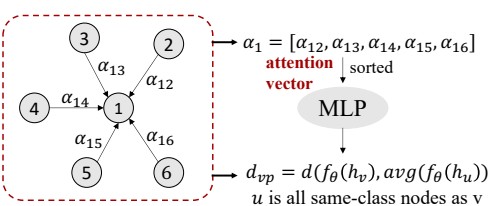

Figure 2: Illustration of distance model

We thus construct an attention vector $\alpha_v$ for each node $v$ to represent the relationship distribution, where a more uniform distribution indicates a higher node importance and a much closer distance to prototype expectation. As shown below and in Figure 2, each component in $\alpha_v$ is an attention score between node $v$ and $u \in N(v)$. Note that a fixed number of neighbors are sampled for each node.

$$\alpha_v = [\alpha_{v1}, \cdots, \alpha_{v|N(v)|}], \tag{13}$$

$$\alpha_{vu} = \frac{exp(\text{LeakyReLU}(\boldsymbol{a}^T[\boldsymbol{W}\boldsymbol{h}_v \parallel \boldsymbol{W}\boldsymbol{h}_u]))}{\sum_{q \in N(v)}(exp(\text{LeakyReLU}(\boldsymbol{a}^T[\boldsymbol{W}\boldsymbol{h}_v \parallel \boldsymbol{W}\boldsymbol{h}_q])))}, \tag{14}$$

where $\boldsymbol{W}$ is a linear transformation, $\parallel$ is a concatenation operation. Attention coefficient is calculated by a single-layer feed-forward neural network with a LeakyReLu nonlinear activation and parameterized by a vector $\boldsymbol{a}$, then a Softmax function is utilized for normalization.

Thus, $\alpha_v$ is the category-irrelevant node representation that describes the relation distribution between given node $v$ and its neighbors. We use sorted $\alpha_v$ as the input of the supervised distance predictor to avoid the effect of neighbor nodes' sampling order. For a node $v$ in class $c$, the distance between node representation and prototype is predicted by a multi-layer supervised model:

$$d(f_\phi(\boldsymbol{x}_v), \mu_c) = MLP(\text{SORTED}(\alpha_v)), \tag{15}$$

where $\boldsymbol{x}_v$ is the node feature, $\mu_c = mean(f_\phi(\boldsymbol{x}_u))$, for all nodes $u$ belongs to class $c$. Then given the support set $\mathbb{S}_c$ of class $c$, the **importance score** $s_v$ is computed by

$$s_v = \frac{exp(-d(f_\phi(\boldsymbol{x}_v), \mu_c))}{\sum_{u \in \mathbb{S}_c} exp(-d(f_\phi(\boldsymbol{x}_u), \mu_c))}. \tag{16}$$

Prototype representation $\overline{\boldsymbol{c}}$ of class $c$ can be obtained by a weighted combination of embeddings,

$$\overline{\boldsymbol{c}} = \sum_{v \in \mathbb{S}_c} s_v f_\theta(\boldsymbol{x}). \tag{17}$$

Then the probability $p(c|v)$ that a node $v$ with feature $\boldsymbol{x}$ belonging to class $c$ can be computed following the Softmax function in Equation( 2). Thus, the loss function $L$ can be defined as a sum over query set $\mathbb{Q}$ of negative log-probability of a node $v$'s true label $c$.

$$L = \frac{1}{N|Q|} \sum_{c=1}^{N} \sum_{v \in \mathbb{Q}_c} -\log p(c|v), \tag{18}$$

where $N$ is the number of classes, $\mathbb{Q}_c$ is the nodes that belong to class $c$ in query set $\mathbb{Q}$. The parameters in representation network $f_\theta(\cdot)$ and importance score network are then updated by SGD.

Table 1: Experiment result on Reddit, Amazon-Electronic and DBLP w.r.t ACC and F1 (%)

| Method | Reddit | | | | Amazon-Electronic | | | | DBLP | | | |
|---|---|---|---|---|---|---|---|---|---|---|---|---|
| | 5-way 3-shot | | 5-way 5-shot | | 5-way 3-shot | | 5-way 5-shot | | 5-way 3-shot | | 5-way 5-shot | |
| | Acc | F1 | Acc | F1 | Acc | F1 | Acc | F1 | Acc | F1 | Acc | F1 |
| Deepwalk | 26.13 | 26.9 | 30.11 | 29.74 | 23.24 | 21.27 | 26.51 | 25.39 | 44.62 | 43.19 | 62.87 | 61.57 |
| node2vec | 27.16 | 26.41 | 31.32 | 29.83 | 24.18 | 24.05 | 27.46 | 26.93 | 41.96 | 40.15 | 58.64 | 58.03 |
| GCN | 38.82 | 38.17 | 45.53 | 44.14 | 54.36 | 54.07 | 56.31 | 55.04 | 58.76 | 57.44 | 69.28 | 69.57 |
| GAT | 39.69 | 39.12 | 47.58 | 46.39 | 53.18 | 52.19 | 56.22 | 56.18 | 60.45 | 59.31 | 70.64 | 68.29 |
| ProtoNet | 34.72 | 33.31 | 37.89 | 38.12 | 53.17 | 52.65 | 58.34 | 57.69 | 37.28 | 36.52 | 43.08 | 44.31 |
| MAML | 29.21 | 26.78 | 32.14 | 33.45 | 54.09 | 54.17 | 59.17 | 58.46 | 38.72 | 39.43 | 45.63 | 44.81 |
| Proto-GCN | 61.43 | 60.74 | 64.08 | 63.15 | 63.18 | 62.49 | 67.32 | 66.93 | 71.35 | 70.72 | 74.89 | 74.66 |
| Meta-GCN | 60.81 | 58.32 | 62.73 | 61.21 | 62.17 | 60.47 | 67.16 | 65.08 | 69.58 | 69.24 | 74.67 | 73.19 |
| Proto-GAT | 61.49 | 60.82 | 63.67 | 64.27 | 63.27 | 62.51 | 68.30 | 67.96 | 71.86 | 70.67 | 74.33 | 73.17 |
| Meta-GAT | 61.45 | 59.34 | 64.06 | 61.48 | 63.86 | 61.55 | 67.97 | 66.74 | 68.62 | 66.94 | 73.38 | 72.17 |
| RALE | 64.73 | 64.42 | 66.35 | 64.27 | 66.82 | 65.39 | 71.48 | 70.05 | 73.17 | 72.46 | 76.95 | 76.47 |
| GPN | 65.37 | 64.28 | 66.57 | 65.19 | 65.69 | 64.32 | 70.31 | 70.24 | 74.69 | 73.82 | 78.58 | 77.43 |
| NIML | **67.51** | **66.90** | **69.67** | **69.33** | **68.93** | **68.24** | **73.85** | **73.19** | **76.53** | **75.32** | **81.37** | **81.09** |

# 6 EXPERIMENT

To verify the effectiveness of NIML on few-shot node classification problem, in this section, we first introduce the experimental settings and then present the detailed experiment results with ablation study and parameter analysis on three public datasets.

## 6.1 EXPERIMENT SETTINGS

We implement the experiment on three public datasets: Reddit (Hamilton et al., 2017), Amazon-Electronic (McAuley et al., 2015), and DBLP (Tang et al., 2008a). Details of datasets are provided in Appendix A.2. $N$ classes are sampled episode by episode from training classes in meta-train phase, and $N$ novel classes from testing classes are used for evaluation. A fixed number of neighbors are sampled to construct the attention vector, where zero is padded for the nodes without enough neighbors. We compare with several baselines which can be grouped into three categories.

- **GNNs**: We test on four graph algorithm including DeepWalk, node2vec, GCN and GAT. Deep-Walk (Perozzi et al., 2014) is done by a series of random work technique, and node embeddings are learnt from the random walks. Node2vec (Grover & Leskovec, 2016) is an extension from DeepWalk, which is a combination of DFS and BFS random walk. GCN (Kipf & Welling, 2016) is like an first-order approximation of spectral graph convolutions. GAT (Veličković et al., 2017) leverages self-attention to enable specifying different weights to different nodes in neighborhood.

- **Meta Learning**: We test on two typical meta learning algorithms without using GNN as backbone. ProtoNet Snell et al. (2017) is a metric-based meta learning method, which learns an embedding function and use prototype to do a classification. MAML Finn et al. (2017) is an optimization-based meta learning method, which learns a good parameter initialization of networks.

- **Meta Learning GNN**: We consider six works that implement GNN in a meta learning framework. Proto-GCN is a baseline we design for an ablation purpose, which learns a GCN as an embedding function and uses the average value as a prototype. Meta-GCN Zhou et al. (2019) is a previous work which extends MAML to graph data by using a GCN base model. Proto-GAT and Meta-GAT are two baselines where the embedding function is GAT. We also include two related works: RALE (Liu et al., 2021) introduces hub nodes and learns both relative and absolute location node embedding; GPN (Ding et al., 2020) learns node importance by aggregating the importance score.

## 6.2 EXPERIMENT RESULTS

Table 1 shows the performance comparison results on 5-way 3-shot and 5-way 5-shot problems on each dataset. We report the average performance of accuracy and F1 score after ten repetitions Among the GNNs, the typical methods DeepWalk and node2vec are far inferior to other methods since they rely on a large number of labeled data to learn good node representations. GCN and GAT

are better than the previous two methods, but they still cannot achieve satisfying performance on this few-shot problem. In terms of ProtoNet and MAML, although they have shown the ability to deal with few-shot problems of Euclidean data, they are hard to handle graph data without considering the graph structure, i.e. node dependency.

Due to the incorporation of both meta-learning and graph structure, the meta-learning GNN model outperforms the previous two types of models, which demonstrates that meta learning methods can effectively deal with the problem of few samples in graph data under a GNN configuration. For the four basic Meta Learning GNN model: Meta-GCN, Proto-GCN, Meta-GAT and Proto-GAT, they all achieve similar performance. Our model NIML outperforms other baselines in each case. The advantage of NIML is slightly advanced in the 5-shot case than in the 3-shot case, thanks to a better refinement of prototype calculation using the importance score in the case of additional nodes.

### 6.3 MODEL ANALYSIS

**Methods of computing importance score**. We implement ablation study to test the performance of different methods of computing importance score and provide results of four models shown in Figure 3. Proto-GCN compute prototype directly by mean function; GPN train a score aggregation model; Proto-GCN+GAT use GAT to learn importance score for each node. The results indicate that distinguishing the importance of various nodes will have a significant impact on the model performance, and NIML is closely connected with the theory conclusion, thus makes its advantages more significant.

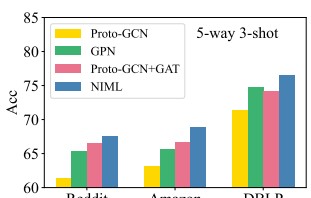

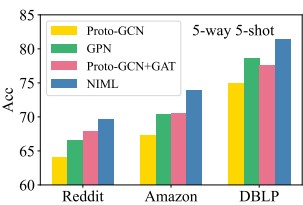

**Effect of $N$-way/ $k$-shot/ $m$-query**. We analyze the effect of number of class $N$, support set size $k$ and query set size $m$ on the accuracy of three datasets. The results of each dataset are depicted in Figure 4. 1) As $N$ grows, the difficulty of predicting increases, resulting in a decline in performance. 2) The accuracy will always increase as $k$ increasing, and the curves tend to flatten in some instances. 3) The query set size $m$ has the least impact on model accuracy of all variables. Larger $m$ may result in decrease in performance, which may be due to the difficulty that larger query sets bring to parameter update.

Figure 3: Different methods of computing importance score

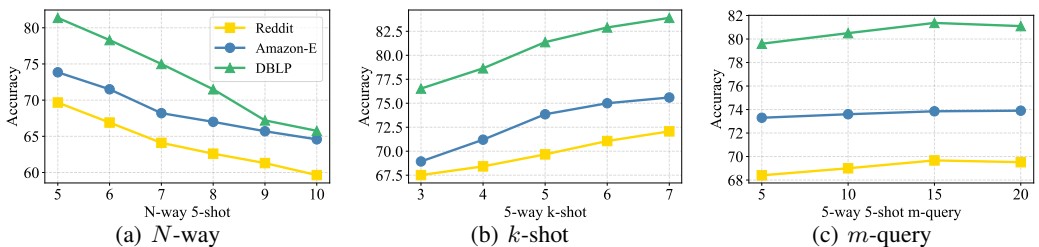

Figure 4: Effect of support set size $k$ on three datasets

### 7 CONCLUSION

This work begins with a theoretical analysis of the effect of node importance on the model, and concludes that providing a greater weight to the data point whose embedding is closer to the expectation of same-class prototype would enhance the lower bound of model accuracy. This theory can also be applied to other domains, not just graph. Then we propose node importance meta learning (NIML) closely based on theoretical conclusion. We construct an attention vector to represent the relationship distribution between node and its neighbors, and train a distance predictor to learn the distance between node embedding and an approximation of prototype expectation. Experiments demonstrate the superior capability of our model in few-shot node classification. NIML has the potential to be utilized in any Proto-based few-shot node classification framework to compute prototype.

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

# A   APPENDIX

## A.1   THEORY PROOF

Table 2: Notation list

| Symbol | Definition | Symbol | Definition |
|--------|-----------|--------|-----------|
| $C$ | Space of classes | $\overline{c_i}$ | Prototype representation in $\mathbb{R}^M$ |
| $\tau$ | Class probability distribution | $f_\phi$ | Embedding Function |
| $\chi$ | Space of input data | $\mu_c$ | Expectation of inputs that belong to class $c$ |
| $N$ | Number of class in a task | $\Sigma_c$ | Expected intra-class variance of class $c$ |
| $\mathbb{S}$ | Support Set | $\Sigma$ | Expected variance between classes |
| $\mathbb{Q}$ | Query Set | $k$ | Number of data points for support set |
| $\mathbb{S}_i$ | Support Set of class $i$ | $m$ | Number of data points for $\mathbb{Q}$ |

### A.1.1   PROOF OF LEMMA 1:

Consider space of classes $C$ with sampling distribution $\tau$, a, b $\overset{\text{iid}}{\sim}$ $\tau$. Let $\mathbb{S} = \{\mathbb{S}_a, \mathbb{S}_b\}$ $\mathbb{S}_a = \{_a\boldsymbol{x}_1, \ldots, _a\boldsymbol{x}_k\}$, $\mathbb{S}_b = \{_b\boldsymbol{x}_1, \ldots, _b\boldsymbol{x}_k\}$, $k \in \mathbb{N}$ is the shot number, and $y(\boldsymbol{x}) = a$. Define $\overline{c_a}$ and $\overline{c_b}$ as shown in Equation( 4). Then,

$$\mathbb{E}_{\boldsymbol{x},\mathbb{S}|a,b}[\alpha] = (\mu_a - \mu_b)^T(\mu_a - \mu_b) + (2\mu_b + \sigma_b - 2\mu_a)^T\sigma_b + \sigma_b^T\sigma_b - \sigma_a^T\sigma_a \tag{19}$$

$$\mathbb{E}_{a,b,\boldsymbol{x},\mathbb{S}}[\alpha] = 2\operatorname{Tr}(\Sigma) + \sigma_b^T\sigma_b - \sigma_a^T\sigma_a \tag{20}$$

$$\mathbb{E}_{a,b}[\operatorname{Var}(\alpha \mid a, b)] \leq 8\left(1 + \frac{1}{k}\right)\operatorname{Tr}\{\Sigma_c\left(\left(1 + \frac{1}{k}\right)\Sigma_c + 2\Sigma\right) + \sigma_b^T\sigma_b + \sigma_a^T\sigma_a\} \tag{21}$$

where

$$\sigma_a = \frac{_aw_{2a}\epsilon_2 - _aw_{1a}\epsilon_1}{_aw_2 + _aw_1}, \sigma_b = \frac{_bw_{2b}\epsilon_2 - _bw_{1b}\epsilon_1}{_bw_2 + _bw_1}$$

**Proof**: From the definition of prototype, we have:

$$\begin{aligned}
\overline{c_a} &= \frac{_aw_1}{_aw_1 + _aw_2} \cdot \phi(_ax_1) + \frac{_aw_2}{_aw_1 + _aw_2} \cdot \phi(_ax_2) \\
&= \frac{_aw_1}{_aw_1 + _aw_2} \cdot (\mu_a - \epsilon_1) + \frac{_aw_2}{_aw_1 + _aw_2} \cdot (\mu_a + \epsilon_2) \\
&= \mu_a + \frac{\epsilon_{2a}w_2 - \epsilon_{1a}w_1}{_aw_1 + _aw_2}
\end{aligned}$$

We denote the second term as $\sigma_a$, thus $\overline{c_a} = \mu_a + \sigma_a$ and $\overline{c_b} = \mu_b + \sigma_b$.

Since $\alpha = \|\phi(x) - \overline{c_b}\|^2 - \|\phi(x) - \overline{c_a}\|^2$,

$$\begin{aligned}
\mathbb{E}_{x,S|a,b}[\alpha] &= \mathbb{E}_{x,S|a,b}[\|\phi(x) - \overline{c_b}\|^2 - \|\phi(x) - \overline{c_a}\|^2] \\
&= \mathbb{E}_{x,S|a,b}[\|\phi(x) - \overline{c_b}\|^2] - \mathbb{E}_{x,S|a,b}[\|\phi(x) - \overline{c_a}\|^2]
\end{aligned}$$

We denote $\mathbb{E}_{x,S|a,b}[\|\phi(x) - \overline{c_b}\|]$ and $\mathbb{E}_{x,S|a,b}[\|\phi(x) - \overline{c_a}\|]$ as $i$ and $ii$ respectively. For a random vector $X$, the expectation of quadratic form is $\mathbb{E}[\|X\|^2] = Tr(Var(X)) + \mathbb{E}^T\mathbb{E}$, thus,

$$\begin{aligned}
i &= \mathbb{E}_{x,S|a,b}[\|\phi(x) - \overline{c_b}\|^2] \\
&= Tr(Var(\phi(x) - \overline{c_b})) + \mathbb{E}[\phi(x) - \overline{c_b}]^T\mathbb{E}[\phi(x) - \overline{c_b}]
\end{aligned}$$

Since $Var(X) = \mathbb{E}[X^2] - (\mathbb{E}[X])^2$,

$$\begin{aligned}
Var(\phi(x) - \overline{c_b}) &= \mathbb{E}[\phi(x) - \overline{c_b}^T(\phi(x) - \overline{c_b})] - \mathbb{E}[\phi(x) - \overline{c_b}]^2 \\
&= \mathbb{E}[\phi(x) - \overline{c_b}^T(\phi(x) - \overline{c_b})] - (\mu_a - \overline{c_b})(\mu_a - \overline{c_b})^T \\
&= \Sigma_c + \mu_a\mu_a^T + \frac{1}{k}\Sigma_c + \overline{c_b c_b}^T - \mu_a\overline{c_b}^T - \overline{c_b}\mu_a^T - [\mu_a\mu_a^T - \mu_a\overline{c_b}^T - \overline{c_b}\mu_a^T + \overline{c_b c_b}^T] \\
&= (1 + \frac{1}{k})\Sigma_c
\end{aligned}$$

Since $\mathbb{E}[\phi(x) - \overline{c_b}] = \mu_a - \overline{c_b}$,

$$i = (1 + \frac{1}{k})\Sigma_c + (\mu_a - \overline{c_b})^T(\mu_a - \overline{c_b})$$

$$ii = (1 + \frac{1}{k})\Sigma_c + (\mu_a - \overline{c_a})^T(\mu_a - \overline{c_a}) = (1 + \frac{1}{k})\Sigma_c + \sigma_a^T\sigma_a$$

Thus,

$$\begin{aligned} i - ii &= (\mu_a - \overline{c_b})^T(\mu_a - \overline{c_b}) - \sigma_a^T\sigma_a \\ &= \mu_a^T\mu_a - \mu_a^T(\mu_b + \sigma_b) - (\mu_b + \sigma_b)^T\mu_a + (\mu_b + \sigma_b)^T(\mu_b + \sigma_b) - \sigma_a^T\sigma_a \\ &= \mu_a^T\mu_a - 2\mu_a^T\mu_b - 2\mu_a^T\sigma_b + \mu_b^T\mu_b + 2\mu_b^T\sigma_b + \sigma_b^T\sigma_b - \sigma_a^T\sigma_a \end{aligned}$$

and,

$$\mathbb{E}_{x,S|a,b}[\alpha] = (\mu_a - \mu_b)^T(\mu_a - \mu_b) + (2\mu_b + \sigma_b - 2\mu_a)^T\sigma_b - \sigma_a^T\sigma_a$$

Since $\mathbb{E}_{a,b,x,S}[\alpha] = \mathbb{E}_{a,b}[\mathbb{E}_{x,S|a,b}[\alpha]]$, we have,

$$\begin{aligned} \mathbb{E}_{a,b,x,S}[\alpha] &= \mathbb{E}_{a,b}[i - ii] \\ &= \mathbb{E}_{a,b}[\mu_a^T - 2\mu_a^T\mu_b + \mu_b^T\mu_b + 2\mu_b^T\sigma_b - 2\mu_a^T\sigma_b + \sigma_b^T\sigma_b - \sigma_a^T\sigma_a] \\ &= Tr(\Sigma) + \mu^T\mu - 2\mu^T\mu + Tr(\Sigma) + \mu^T\mu + 2\mu^T\sigma_b - 2\mu_T\sigma_b + \sigma_b^T\sigma_b - \sigma_a^T\sigma_a \\ &= 2Tr(\Sigma) + \sigma_b^T\sigma_b - \sigma_a^T\sigma_a \end{aligned}$$

Thus, $\mathbb{E}_{a,b,x,S}[\alpha] = 2Tr(\Sigma) + \sigma_b^T\sigma_b - \sigma_a^T\sigma_a$.

Then we do an inequality scaling on the variance of $\alpha$.

$$\begin{aligned} Var(\alpha|a,b) &= Var(\|\phi(x) - \overline{c_b}\|^2 - \|\phi(x) - \overline{c_a}\|^2) \\ &= Var(\|\phi(x) - \overline{c_b}\|^2) + Var(\|\phi(x) - \overline{c_a}\|^2) - 2Cov(\|\phi(x) - \overline{c_b}\|^2, \|\phi(x) - \overline{c_a}\|^2) \\ &\leq Var(\|\phi(x) - \overline{c_b}\|^2) + Var(\|\phi(x) - \overline{c_a}\|^2) + 2\sqrt{Var(\|\phi(x) - \overline{c_b}\|^2)Var(\|\phi(x) - \overline{c_a}\|^2)} \\ &\leq 2Var(\|\phi(x) - \overline{c_b}\|^2) + 2Var(\|\phi(x) - \overline{c_a}\|^2) \end{aligned}$$

Given the theorem: given a random vector $y\ N(\mu, \Sigma)$, $A$ is a symmetric matrix,

$$Var(y^TAy) = 2Tr((A\Sigma)^2) + 4\mu^TA\Sigma A\mu$$

we can obtain that,

$$Var(\|\phi(x) - \overline{c_b}\|^2) = 2(1 + \frac{1}{k})^2Tr(\Sigma_c^2) + 4(1 + \frac{1}{k})(\mu_a - \overline{c_b})^T\Sigma_c(\mu_a - \overline{c_b})$$

$$Var(\|\phi(x) - \overline{c_a}\|^2) = 2(1 + \frac{1}{k})^2Tr(\Sigma_c^2) + 4(1 + \frac{1}{k})\sigma_a^T\Sigma_c\sigma_a$$

Thus,

$$\begin{aligned} \mathbb{E}_{a,b}[Var(\alpha|a,b)] &\leq \mathbb{E}_{a,b}[2Var(\|\phi(x) - \overline{c_b}\|^2) + 2Var(\|\phi(x) - \overline{c_a}\|^2)] \\ &= \mathbb{E}_{a,b}[8(1 + \frac{1}{k})^2Tr(\Sigma_c^2) + 8(1 + \frac{1}{k})[(\mu_a - \overline{c_b})_c^T(\mu_a - \overline{c_b}) + \sigma_a^T\Sigma_c\sigma_a]] \\ &= 8(1 + \frac{1}{k})\mathbb{E}_{a,b}[Tr\{(1 + \frac{1}{k})\Sigma_c^2 + \Sigma_c((\mu_a - \overline{c_b})^T(\mu_a - \overline{c_b}) + \sigma_a^T\sigma_a)\}] \\ &= 8(1 + \frac{1}{k})Tr\{\Sigma_c[(1 + \frac{1}{k})\Sigma_c + 2\Sigma + \sigma_b^T\sigma_b + \sigma_a^T\sigma_a]\} \end{aligned}$$

### A.1.2 PROOF OF THEOREM 1

Under the condition where Lemma 1 hold, we have:

$$R(\phi) \geqslant \frac{(2\,Tr(\Sigma) + \sigma_b^T\sigma_b - \sigma_a^T\sigma_a)^2}{f_1(\sigma_a, \sigma_b) + f_2(\sigma_a, \sigma_b)} \tag{22}$$

where

$$f_1(\sigma_a, \sigma_b) = 12 \operatorname{Tr}\{\Sigma_c(\frac{3}{2}\Sigma_c + 2\Sigma + \sigma_b^T\sigma_b + \sigma_a^T\sigma_a)\}$$

$$f_2(\sigma_a, \sigma_b) = \mathbb{E}_{a,b}[((\mu_a - \mu_b)^T(\mu_a - \mu_b) + (2\mu_b + \sigma_b)^T\sigma_b)^2]$$

**Proof:** From the three equations in Lemma 1, we plug in the result to Equation(7) and do an inequality scaling as shown in below. Since we know:

$$
\begin{aligned}
Var(\alpha) &= \mathbb{E}[\alpha^2] - \mathbb{E}[\alpha]^2 \\
&= \mathbb{E}_{a,b|x,S}[\alpha^2|a,b] - \mathbb{E}_{a,b,x,S}[\alpha]^2 \\
&= \mathbb{E}_{a,b}[Var(\alpha|a,b) + \mathbb{E}_{x,S}[\alpha|a,b]^2] - \mathbb{E}_{a,b,x,S}[\alpha]^2
\end{aligned}
$$

Then,

$$
\begin{aligned}
R(\phi) &\geq \frac{2Tr(\Sigma) + \sigma_b^T\sigma_b - \sigma_a^T\sigma_a}{f_1(\sigma_a,\sigma_b) + \mathbb{E}_{a,b}[[(\mu_a - \mu_b)^T(\mu_a - \mu_b) + (2\mu_b + \sigma_b - 2\mu_a)^T\sigma_b - \sigma_a^T\sigma_a]^2]} \\
&\geq \frac{2Tr(\Sigma) + \sigma_b^T\sigma_b - \sigma_a^T\sigma_a}{f_1(\sigma_a,\sigma_b) + f_2(\sigma_a,\sigma_b)}
\end{aligned}
$$

where

$$f_1(\sigma_a, \sigma_b) = 8(1 + \frac{1}{k})\operatorname{Tr}\{\Sigma_c(\left(1 + \frac{1}{k}\right)\Sigma_c + 2\Sigma + \sigma_b^T\sigma_b + \sigma_a^T\sigma_a)\}$$

$$f_2(\sigma_a, \sigma_b) = \mathbb{E}_{a,b}[((\mu_a - \mu_b)^T(\mu_a - \mu_b) + (2\mu_b + \sigma_b)^T\sigma_b)^2]$$

In the 2-way 2-shot case we talked about, $k = 2$.

### A.1.3 EXTEND THE ALGORITHM TO N CLASS

Let **x** and $y$ denote the pair of query set. Let $\alpha_i = \|\phi(x) - \overline{c_i}\|^2 - \|\phi(x) - \overline{c_y}\|^2$, hence $R(\phi) = Pr_{c,x,S}(\cup_{i=1,i\neq y}^N \alpha_i > 0)$.

By Frechet's inequality:

$$R(\phi) > \sum_{i=1,i\neq n}^N Pr(\alpha_i > 0) - (N - 2)$$

After plug in the inequality of $R(\phi)$ in Theorem 1, the lower bound of accuracy for N classes problem can be obtained.

### A.2 EXPERIMENT DETAILS

### A.2.1 DATASET DESCRIPTION

Table 3: Statistic of datasets

| Dataset | # of nodes | # of edges | # of features | labels | train/val/test split |
|---|---|---|---|---|---|
| Reddit | 232,965 | 11,606,919 | 602 | 41 | 16/10/15 |
| Amazon-Electronic | 42,318 | 43,556 | 8,669 | 167 | 90/37/40 |
| DBLP | 40,672 | 288,270 | 7,202 | 137 | 80/27/30 |

**Reddit** (Hamilton et al., 2017) is a social network with data sampled from Reddit, where each node is a discussion post and an edge between two nodes means that the two posts are commented by the same user.

**Amazon-Electronic** (McAuley et al., 2015) is a product network within electronic category of Amazon. Nodes represent products, and edges between two products exits if they are bought together.

**DBLP** (Tang et al., 2008a) is a citation network where each node is a paper and link is the citation relationship between papers.

We record the number of nodes contained in each category in these three datasets and show the results of Reddit dataset in the histogram.

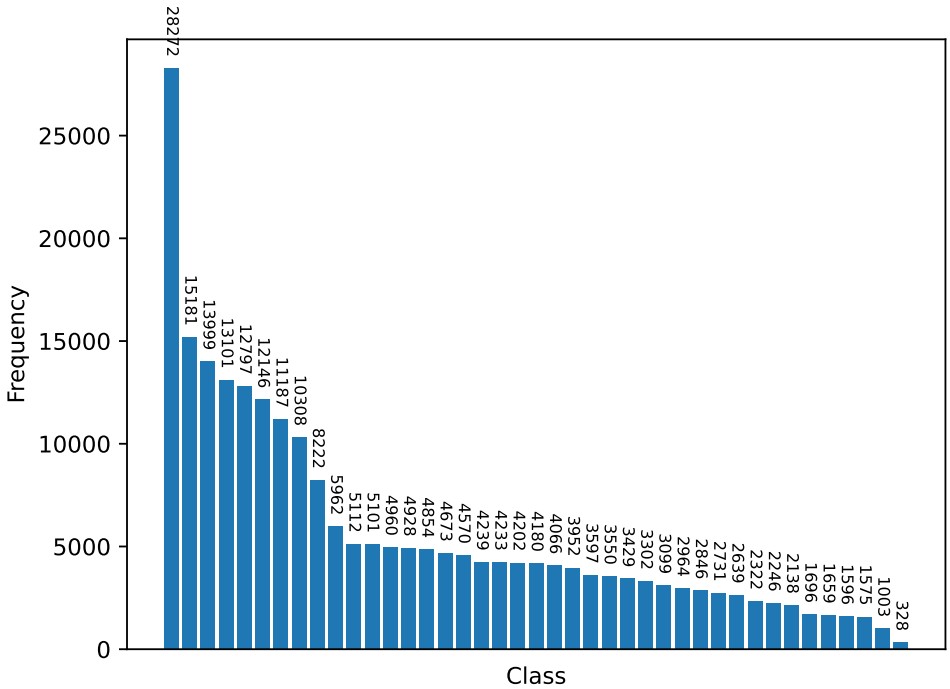

Figure 5: Histogram of Reddit dataset

### A.2.2 IMPLEMENTATION DETAILS

We implement the proposed framework in PyTorch. We set the number of episode as 500 with an early stopping strategy. The representation network $f_\theta(\cdot)$, i.e. GCN, consists of two layers with dimension size 32 and 16, respectively. Both of them are activated with ReLU function. We train the model using Adam optimizer, whose learning rate is set to be 0.005 initially with a weight decay of 0.0005. The size of query set is set to be 15 for all datasets. The Proto-GCN and distance predictor are both learnt during meta-train phase. We also provide an anonymous Github link in the supplementary file.

### A.3 TECHNICAL EXPLANATION

Figure 6 provides an illustration of difference between the Proto-based GCN and distance predictor, where the bottom right figure depicts the embedding space of a prototypical network and the upper right figure is the distance in the embedding space between a given node and its same-class prototype. The distance is equivalent to the length of gray arrow in bottom right figure.

### A.4 DIFFERENCE BETWEEN NIML AND GPN

Even though, both NIML and GPN make an effort to compute weighted prototypes, the two methods are designed with different intentions. NIML starts with a theoretical analysis, quantify the node importance as the distance from the node to its same-class prototype expectation and conclude that assigning higher weights to nodes with closer distance will enhance the lower bound of model accuracy. After that, NIML adopts the idea that the distribution of the relationship between a given node and its neighbors can reflect the node importance and then construct an attention vector that depicts the relationship distribution as input to predict the distance in a supervised manner, further learning the node importance. While GPN adopts a different view that assumes the importance of a node is highly correlated with its neighbor's importance and derive a score aggregation mechanism

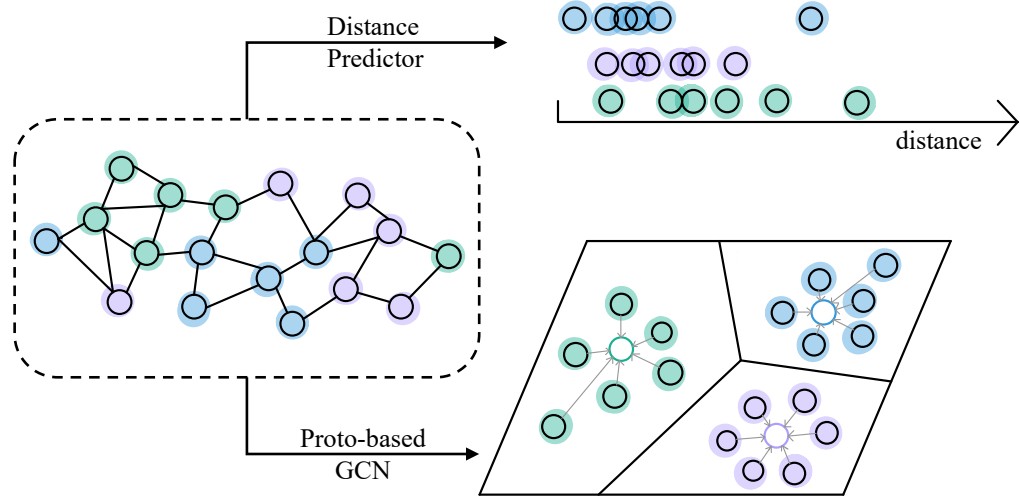

Figure 6: Difference between Proto-based GCN model and distance predictor: distance predictor maps node of similar importance to similar distance value regardless of node category, Proto-based GCN maps same-class nodes to close locations in the embedding space.

using GAT as the backbone, which has similar characteristic to message passing that relies on graph homophily. We think this is the main reason why NIML outperforms GPN as shown in Table 1.

### A.5 VISUALIZATION OF RELATIONSHIP BETWEEN SCORE AND DISTANCE

In order to verify whether NIML follows the theory, we visualize the relationship between score and distance in figure 7. For a selected category, we calculate the embedding of five nodes with the same label belonging to the support set and visualize them in the figure together with the prototype expectation (mean of all same-class embeddings) of that category. The shade of the color represents the score. The darker the color, the higher the score, where the darkest color is the prototype. The distance between points in the figure is consistent with the distance between node embedding. Here we present three groups of visualization. From the result, we find that our algorithm always assigns higher weights to closer nodes, but very strict distinctions may not be made for certain cases where the distance is relatively close. Although the detail of some cases is inconsistent, the overall trend is consistent with the theory.

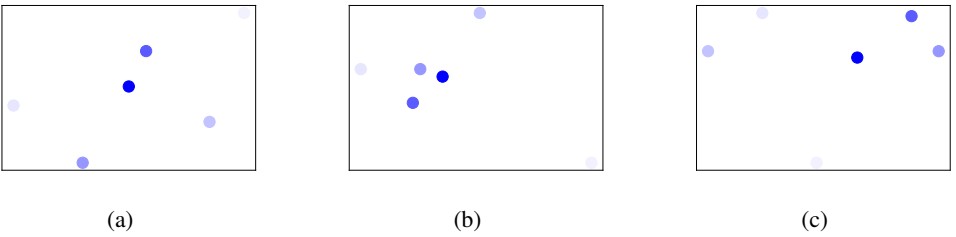

|  (a) | (b) | (c) |

Figure 7: Visualization of the relationship between score and distance

