# OpenReview forum: "Node Importance Specific Meta Learning in Graph Neural Networks"
_ICLR.cc/2023/Conference — Submitted to ICLR 2023_

### Official Review · Reviewer_meJA · 2022-10-24

**Confidence:** 4
**Correctness:** 3
**Technical Novelty And Significance:** 2
**Empirical Novelty And Significance:** 2
**Recommendation:** 5

**Clarity, Quality, Novelty And Reproducibility:**

Generally speaking, this work is well-organized and easy to understand. However, due to the limited novelty of the model design, the gaps between the theoretical proof and the model design, and some presentation mistakes, I would suggest that the model design should be improved and some explanation about the model design should be clearly clarified.

**Strength And Weaknesses:**

Strength:

1. This work provides detailed theoretical proof to demonstrate that assigning higher weights to the samples that have closer distances to the prototype expectation will increase the lower bound of the model accuracy.

2. In general, this manuscript is well written with clear motivations, detailed related works, rigorous proofs, and relatively sufficient experiments, which is easy to follow.

Weakness:

1. There are some gaps between the theoretical analysis and the model design. The theoretical conclusion is that higher weights of samples will increase the lower bound of the model performance, which demonstrates that the model design of existing work Proto-GCN and the proposed model NIML are solid. This work proposes the category-irrelevant node attentions and further designs the weighted prototype based on the importance scores. However, the motivations of the proposed category-irrelevant node attentions (different from Proto-GCN) and the weighted prototype are not clearly discussed in this work.

2. It claims that Proto-GCN as the input does not meet the expectations for distance predictor. But it does not provide a detailed explanation. Why is it necessary to design a category-irrelevant node information input? Why Proto-GCN is not suitable for your framework? What’s the main contribution of your model design?

3. The model design is rather simple and not novel enough. The main idea of the model design is to calculate the attention scores between nodes and further learn a weighted prototype. The theoretical analysis is to prove that node importance can improve the lower bound of the model performance. From my perspective, there are some gaps between the model design and the theoretical proof. You should provide the theoretical analysis of your model design and validate that your model design is better than Proto-GCN.

4. The discussion of related works is comprehensive, but some discussion about related works is a little bit vague and does not show the key idea of the related works. For instance, in section 2.3, “MetaHG presents a heterogeneous graph learning model for automatically detecting illicit drug traffickers on Instagram.” The main idea of this work is to design a meta-learning model on heterogeneous graphs to address the problem of the limited labeled drug traffickers on Instagram.

5. The presentations of most equations need to be revised. For instance, there should have a comma between the equation and ‘where’ (e.g., equations 3, 4,6, and 10). There should have a period after the equation (e.g., equations 1,2,5).

**Summary Of The Paper:**

This manuscript proposes NIML to investigate the node importance in node classification meta-learning tasks. Specifically, it theoretically demonstrates the node importance between neighbors can increase the lower bound of the model accuracy. Then it proposes the node importance meta-learning architecture to learn the importance score of each node and further train a weighted prototype for meta-learning. Extensive experiments on three benchmark datasets validate the superiority of NIML.

**Summary Of The Review:**

This manuscript is well-written and easy to follow. Besides, it provides a detailed theoretical analysis of the influence of node weights on model performance, which is impressive. However, due to the limited novelty of the model design, gaps between theoretical proof and the model design, and unclear clarification of the model, I would suggest this paper further enhance the model design.

---

> ### Author Response · Authors · 2022-11-17
> **Response to Reviewer meJA 1/2**
>
> We thank Reviewer meJA for the insightful comments as well as for acknowledging our theoretical analysis. We address the concerns and questions as follows.
>
> > **Weakness 1**: There are some gaps between the theoretical analysis and the model design. The theoretical conclusion is that higher weights of samples will increase the lower bound of the model performance, which demonstrates that the model design of existing work Proto-GCN and the proposed model NIML are solid. This work proposes the category-irrelevant node attentions and further designs the weighted prototype based on the importance scores. However, the motivations of the proposed category-irrelevant node attentions (different from Proto-GCN) and the weighted prototype are not clearly discussed in this work.
> >
>
> In our paper, Proto-GCN is used to refer to a ProtoNet using GCN as the embedding function, and we guess you mean to refer to the existing work that studies node importance: GPN[1].
>
> Please check the summary rebuttal post that addresses common issues, wherein in the first part we discuss the significance of exploring the node importance in graph few-shot problems and the difference between NIML and GPN[1]; and in the second part, we describe the model design motivation and how the model is closely combined with the theoretical results. We hope the description of the idea of model design and how the model is closely integrated with the theory can answer your doubts.
>
> For the motivation of the proposed category-irrelevant node attentions, we think the word "category-irrelevant" we used could not accurately convey the feature of what we want the distance prediction model input to be, and thus it makes you confused. Actually, the attention vector is not totally irrelevant to the category since the process we compute attention involves node features that are still category-relevant. We have changed the confusing description in the revised version and explain why we choose to construct an attention vector as the input below.
>
> For the distance prediction model, we hope that the input reflects as little label information as possible. As shown in Figure 5 in the appendix, in the bottom right subfigure, we illustrate the embedding space that Proto-based GCN learns for the classification purpose, where each node (solid circle) is close to the prototype (hollow circle) of the category it belongs to and far from the prototypes of other categories. Then in the upper right subfigure, we present the one-dimensional distance space that shows the distance between a given node and its same-class prototype, the distance, as discussed in the theoretical part, represents the node importance. Thus, we can notice the fact that different from embedding space generated from Proto-based GCN, nodes belonging to different categories can be mapped to the same value in distance space. This means that if we would like to accurately learn the distance, we should not directly use the node feature as the distance predictor input, which includes much information about the node's category, as Proto-based GCN did. The reason why we cannot directly use node feature or node embedding learned from Proto-GCN is that they both reflect more label information rather than importance information.
>
> We need input to truly reflect the importance of a node in its category and avoid the interference caused by the category information of a node as much as possible. We thus resort to the idea that the distribution of relationships between a given node and its neighbor can reflect the importance of a node, and construct an attention vector as the model input to represent the relationship distribution.

---

> > ### Author Response · Authors · 2022-11-17
> > **Response to Reviewer meJA 2/2**
> >
> > > **Weakness 2**: It claims that Proto-GCN as the input does not meet the expectations for distance predictor. But it does not provide a detailed explanation. Why is it necessary to design a category-irrelevant node information input? Why Proto-GCN is not suitable for your framework? What is the main contribution of your model design?
> > >
> >
> > We have answered the first two questions in the response to weakness 1, and we hope that can solve your confusion. We restate our contribution to model design as follows.
> >
> > Our model design is fully motivated by the theoretical analysis that assigning higher weights to nodes that have a closer distance to the same-class prototype expectation will improve the model performance. As the prototype expectation can be approximated by the mean value of all same-class nodes involved in the training process, the ground truth: distance between a node embedding and its same-class prototype can be obtained. Then we design the input: an attention vector, to describe the relationship distribution between a given node and its neighbors. Finally, we predict the distance for each node in the test phase and assign weight (when computing the prototype) according to the distance value to make sure the method follows the theoretical conclusion.
> >
> > > **Weakness 3**: The model design is rather simple and not novel enough. The main idea of the model design is to calculate the attention scores between nodes and further learn a weighted prototype. The theoretical analysis is to prove that node importance can improve the lower bound of the model performance. From my perspective, there are some gaps between the model design and the theoretical proof. You should provide the theoretical analysis of your model design and validate that your model design is better than Proto-GCN.
> > >
> >
> > We have shown the close relationship between our model design and theoretical proof in the response to weakness 1 and weakness 2. We briefly restate that the theory part gives us guidance that assigning higher weights to nodes that have closer distance will enhance the lower bound of model accuracy. Thus we design a distance predictor to directly predict the distance for each node in the test phase and follows the strategy in theory to assign weights.
> >
> > > **Weakness 4**: The discussion of related works is comprehensive, but some discussion about related works is a little bit vague and does not show the key idea of the related works. For instance, in section 2.3, "MetaHG presents a heterogeneous graph learning model for automatically detecting illicit drug traffickers on Instagram." The main idea of this work is to design a meta-learning model on heterogeneous graphs to address the problem of the limited labeled drug traffickers on Instagram.
> > >
> >
> > Thanks for recognizing the related works section. We are sorry for not mentioning the detail that MetaHG[2] is a work focused on the few-shot problem, we have updated the description in the revised version, and thank you again for the advice.
> >
> > > **Weakness 5**: The presentations of most equations need to be revised. For instance, there should have a comma between the equation and where (e.g., equations 3, 4,6, and 10). There should have a period after the equation (e.g., equations 1,2,5).
> > >
> >
> > Thanks for pointing out the problem with the equation format, we have revised the format in the newest version.
> >
> > **Reference**
> >
> > [1] Ding, Kaize, et al. "Graph prototypical networks for few-shot learning on attributed networks."*Proceedings of the 29th ACM International Conference on Information & Knowledge Management*. 2020.
> >
> > [2] Qian, Yiyue, et al. "Distilling Meta Knowledge on Heterogeneous Graph for Illicit Drug Trafficker Detection on Social Media."*Advances in Neural Information Processing Systems* 34 (2021): 26911-26923.

---

> ### Comment · Reviewer_meJA · 2022-11-26
> **Reply to authors' response**
>
> Dear authors,
>
> Thank you for your response. I have also read all the other reviews. With further explanation, it is easier for me to understand the relationship between the theoretical proof and the model design. However, my major concern about the incremental novelty still exists. The model design is a simple combination of existing methods, i.e., attention mechanism to calculate the node importance, and the weighted prototype based on the node embeddings, which is incremental to me. Unfortunately, I think the current work is not good enough for ICLR and I keep my score.

---

> > ### Author Response · Authors · 2022-12-09
> > **Thanks for your comments**
> >
> > Thank you again for your response and constructive comments. We understand that novelty can be subjective. The novelty of our approach lies in its logic and close relationship with theoretical analysis, that is to construct a distribution representing the community of nodes as the input to the distance predictor, and use the learned distance/score to refine the prototype embeddings. The attention mechanism is not used directly to represent the node importance, it is the method to construct the relationship distribution (input to the distance predictor).

---

### Official Review · Reviewer_aDYS · 2022-10-27

**Confidence:** 4
**Clarity, Quality, Novelty And Reproducibility:** The paper is well-written and easy to…
**Correctness:** 3
**Technical Novelty And Significance:** 2
**Empirical Novelty And Significance:** 2
**Recommendation:** 5

**Strength And Weaknesses:**

Strengths:

1. Few-shot learning on graphs is an important and hot topic.

2. Theoretical evidence is welcomed to verify the necessity of considering the node importance in the calculation of the prototype.


Weaknesses:

1. One of my main concerns is that, the idea of considering node importance in each task is not novel. The authors also mentioned that some work (e.g., Ding et al., 2020) has been proposed based on this point for few-shot classification. I feel this limits the contribution of this paper. Though theoretical analysis for this point is provided, in my opinion, only theoretical analysis is not sufficient enough for a paper. A better organizational form of a paper is to propose a novel/interesting model which is associated with the corresponding theoretical analysis.

2. For the calculation of neighborhood weights, why need to fix the number of neighbors? Is it possible to use all neighbors, since neighborhood sampling may result in information loss?

3. Eq.(15) does not employ $\mu_c$ as input. So why it can calculate the distance between node $v$ and center $c$?

4. From my view, I feel the experiments are not quite sufficient. It is better to provide more model analysis for demonstration.

**Summary Of The Paper:**

This paper investigates the problem of few-shot node classification on graphs. To improve the performance, the authors find that the node importance in each task is very important and should be taken into consideration when generating the prototype of each class. In particular, the authors theoretically and empirically demonstrate their viewpoints.


**Summary Of The Review:**

In summary, this paper provides the theoretical analysis to an existing simple approach, which I think is not quite sufficient enough, and not quite novel. The experiments are not sufficient, either.

---

> ### Author Response · Authors · 2022-11-17
> **Response to Reviewer aDYS 1/2**
>
> We thank reviewer aDYS for the constructive comments. We address the problems and concerns as follows.
>
> > **Weakness 1**: One of my main concerns is that, the idea of considering node importance in each task is not novel. The authors also mentioned that some work (e.g., Ding et al., 2020) has been proposed based on this point for few-shot classification. I feel this limits the contribution of this paper. Though theoretical analysis for this point is provided, in my opinion, only theoretical analysis is not sufficient enough for a paper. A better organizational form of a paper is to propose a novel/interesting model which is associated with the corresponding theoretical analysis.
> **Summary of the review**: In summary, this paper provides the theoretical analysis to an existing simple approach, which I think is not quite sufficient enough, and not quite novel.
> >
>
> Please check the summary rebuttal post that addresses common issues, wherein in the first part we discuss the significance of exploring the node importance in graph few-shot problems and the difference between NIML and GPN[1]; and in the second part, we describe the model design motivation and how the model is closely combined with the theoretical results.
>
> We agree with you that a great paper should propose an interesting model which is associated with the theoretical analysis. We would like to emphasize that our contribution is not only to give a theoretical analysis of the node importance problem but also to propose the distance predictor based on the theoretical results as discussed in the summary post. Besides, the theoretical results in our paper cannot provide support for the previous work GPN since the key component: the distance between a node embedding and its same-class prototype expectation does not have a corresponding part in the score aggregation method proposed by GPN.
>
> > **Weakness 2**: For the calculation of neighborhood weights, why need to fix the number of neighbors? Is it possible to use all neighbors, since neighborhood sampling may result in information loss?
> >
>
> The reason we need to fix the number of neighbors is that we train the distance predictor in a supervised manner. The input is the attention vector $\alpha_v = [\alpha_{v1},\dots,\alpha_{|N(v)|}]$ for a given node $v$, and the output is the distance between node embedding $f_{\phi}(x_v)$ and the same-class prototype embedding $\mu_c$. The ground truth of distance is known when training the distance predictor. We need to fix the number of neighbors since we need to keep the dimensionality of the model input $\alpha_v$ invariant.
>
> As you point out that neighbor sampling may result in information loss, we actually use almost all neighbors for each node. For nodes with insufficient number of neighbors, we pad their attention vector with zero. Thus, under the requirement of fixing the input length, we preserve the maximum amount of neighbor relationship information, as well as including information about the number of neighbors.
>
> > **Weakness 3**: Eq.(15) does not employ $\mu_c$ as input. So why it can calculate the distance between node $v$ and center $c$?
> >
>
> Eq.(15) is the distance predictor model with attention vector $\alpha_v$ as input and the distance as output. The design of this model is based on the idea that the distribution of relationships between a node and its neighbor can reflect the distance from the node to the same-class prototype expectation. Therefore, in Eq. (15), given a node's attention vector (i.e. neighbor relationship distribution), the distance from the node to the prototype expectation can be predicted.
>
> We cannot include the prototype expectation $\mu_c$ as input, because $\mu_c$ can only be calculated during the training process when we get access to relatively abundant same-class nodes; while in the test process, we only have $k$ nodes for each category in support set, which means that $\mu_c$ can not be calculated.

---

> > ### Author Response · Authors · 2022-11-17
> > **Response to Reviewer aDYS 2/2**
> >
> > > **Weakness 4**: From my view, I feel the experiments are not quite sufficient. It is better to provide more model analysis for demonstration.
> > >
> >
> > Thanks for your suggestion of adding more experiments. From the theoretical conclusion, we know that assigning higher weights to nodes that have a closer distance to the expected prototype embedding will enhance the lower bound of accuracy. Thus we train a **general distance predictor** in a supervised manner in order to tightly connect the theory with the methodology.
> >
> > To demonstrate the generality and efficiency of NIML, we choose a graph few-shot work: G-Meta[1], and apply NIML to the prototype computation part. G-Meta uses a local subgraph to represent each node and adopts the same framework to different scenarios including single graph with disjoint labels, multiple graphs with shared labels, and multiple graphs with disjoint labels.
> >
> > We apply the distance predictor of NIML on the prototype computation part in G-Meta and present the results in three scenarios below. obgn-arxiv is the scenario: single graph with disjoint labels; Tissue-PPI is the scenario: multiple graphs with shared labels; Fold-PPI is the scenario: multiple graphs with disjoint labels.
> >
> > |  | obgn-arxiv | Tissue-PPI | Fold-PPI |
> > | --- | --- | --- | --- |
> > | G-Meta | 0.442 | 0.713 | 0.502 |
> > | G-Meta+NIML | 0.491 | 0.742 | 0.534 |
> >
> > The above results clearly show that NIML strengthens G-Meta in all three scenarios, which shows the high efficiency and good adaptability of NIML. The magnitude of the improvement is greatest in the case of a single graph. The reason might be that in the case of a single graph, the distribution of relationships represented by each node's attention vector contains more similar information.
> >
> > **Reference**
> >
> > [1] Ding, Kaize, et al. "Graph prototypical networks for few-shot learning on attributed networks."*Proceedings of the 29th ACM International Conference on Information & Knowledge Management*. 2020.

---

### Official Review · Reviewer_sckx · 2022-10-30

**Confidence:** 4
**Clarity, Quality, Novelty And Reproducibility:** Overall clarity is good. Idea is nove…
**Correctness:** 3
**Technical Novelty And Significance:** 3
**Empirical Novelty And Significance:** 3
**Recommendation:** 6

**Strength And Weaknesses:**

Strengths:
S1. The paper is well balanced between model design, theoretical analysis and empirical results.

S2. The paper is well written and well motivated.

S3. The empirical results are promising.

Weaknesses:
W1. The theoretical analysis is based on the lower bound of the accuracy, which shows that a weighted strategy can increase the lower bound. However, there is still a gap between the lower bound and the true accuracy. Without analysing how tight the bound is, an increase in the lower bound may not necessarily imply the impact on the accuracy. Some discussion on the tightness of the bound can be beneficial.

W2. Ablation study on the importance score: Compared to Proto-GCN, Proto-GCN+GAT can already improve the performance quite a lot, and the marginal benefit brought by the proposed method NIML is relatively small (note the y axis starts from 60. So do we really need the proposed importance calculation? What is the computational overhead compared to just using attention as the importance score?

**Summary Of The Paper:**

This paper proposes a meta-learning approach for few shot node classification on a graph. In particular, a ProtoNet metric-based approach is adopted. However, unlike traditional ProtoNet where the class prototypes are computed from class samples in an unweighted manner, this paper assigns different importance scores to different samples. The importance scores are derived from a distance prediction function, which is in turn based on an attention mechanism. The paper also analysed the theoretical properties, in which a weighted scheme can improve the lower bound on the accuracy, compared to the unweighted scheme. Experiments are conducted to evaluate the efficacy of the proposed approach.

**Summary Of The Review:**

Overall, I think this is a good paper where the strengths overweigh the weaknesses.

---

> ### Author Response · Authors · 2022-11-17
> **Response to Reviewer sckx**
>
> We thank reviewer sckx for the constructive comments to help us improve the paper. We address the problems and concerns as follows.
>
> > **Weakness 1**: The theoretical analysis is based on the lower bound of the accuracy, which shows that a weighted strategy can increase the lower bound. However, there is still a gap between the lower bound and the true accuracy. Without analysing how tight the bound is, an increase in the lower bound may not necessarily imply the impact on the accuracy. Some discussion on the tightness of the bound can be beneficial.
> >
>
> Thanks for your suggestion. We analyze the inequality scaling to investigate how tight the lower bound is. During the proof, we perform inequality scaling at two places in total. The first one is in the proof of Lemma 1, where we do an inequality scaling on the variance of $\alpha$ as follows.
>
> $$
> Var(\alpha|a,b) = Var(\left \| \phi(x) - \overline{c_b}\right\|^2 - \left \| \phi(x) - \overline{c_a}\right\|^2)) \\
> \leq 2Var(\left \| \phi(x) - \overline{c_b}\right\|^2) + 2Var(\left \| \phi(x) - \overline{c_a}\right\|^2)
> $$
>
> The second one is in the proof of Theorem 1, where we do an inequality scaling in one term of the denominator as follows.
>
> $$
> \mathbb{E}[[(\mu_a-\mu_b)^T(\mu_a-\mu_b) + (2\mu_b+\sigma_b - 2\mu_a)^T\sigma_b - \sigma_a^T\sigma_a]^2] \\
> \leq \mathbb{E}[((\mu_a-\mu_b)^T(\mu_a-\mu_b) + (2\mu_b+\sigma_b)^T\sigma_b)^2]
> $$
>
> As we discussed in the theoretical conclusion, when $\sigma_a$ and $\sigma_b$ are close to 0, the lower bound will be improved. The two components which we omit when doing the inequality scaling are both close to 0 when $\sigma_a$ and $\sigma_b$ are close to 0, which means the bound is very tight.
>
> > **Weakness 2**: Ablation study on the importance score: Compared to Proto-GCN, Proto-GCN+GAT can already improve the performance quite a lot, and the marginal benefit brought by the proposed method NIML is relatively small (note the y axis starts from 60. So do we really need the proposed importance calculation? What is the computational overhead compared to just using attention as the importance score?
> >
>
> We first would like to claim that in Proto-GCN+GAT, we do not directly use attention as the importance score since attention can only depict the relationship between the node and its neighbors, but we need the importance score to describe the important degree of the node among all the same-class nodes. Actually, in Proto-GCN+GAT, we adopt a trainable GAT framework that uses a linear layer as the final output layer to output a score for each node. Thus, we need to backpropagate through the GAT to learn the parameters at each iteration. While in NIML, the update of embedding function GCN and the distance prediction are independent part, that is, linear relationship. Thus, the computational cost of NIML is much smaller than that of Proto-GCN+GAT.
>
> As for the benefit brought by NIML compared to Proto-GCN+GAT, we do not think that NIML always has marginal benefit. For example, under the 5-way 5-shot setting, in both Amazon-E and DBLP datasets, the improvement of NIML over Proto-GCN is twice that of Proto-GCN+GAT over Proto-GCN. Besides, among the three methods of estimating node importance, neither GPN[1] nor Proto-GCN+GAT can outperform the other in all situations and all datasets, but NIML outperforms the other two methods in all settings, which shows the effectiveness of NIML.
>
> **Reference**
>
> [1] Ding, Kaize, et al. "Graph prototypical networks for few-shot learning on attributed networks."*Proceedings of the 29th ACM International Conference on Information & Knowledge Management*. 2020.

---

### Official Review · Reviewer_gYK6 · 2022-11-03

**Confidence:** 4
**Correctness:** 3
**Technical Novelty And Significance:** 1
**Empirical Novelty And Significance:** 2
**Recommendation:** 3

**Clarity, Quality, Novelty And Reproducibility:**

Clarity, Quality: The paper is clear and easy to follow.

Novelty: I think the paper is lack of novelty as the method relevant to node importance has already been used in existing works. I do not think a theoretical analysis is novel enough.

Reproducibility: I think most of the experiment results in the paper can be reproduced.

**Strength And Weaknesses:**

Strength:

1.The paper theoretically searches for the node importance in few-shot graph node classification problems and demonstrates that the node importance can help the model get a higher accuracy lower bound.

2.The experiment shows that the proposed NIML performs better than other baselines.

Weakness:

1.I think the work is lack of novelty as the work GPN[1] has already proposed to add node importance score in the calculation of class prototype and the paper only give a theoretical analysis on it.

2.The experiment part is not sufficient enough.
(1) For few-shot graph node classification problem to predict nodes with novel labels, there are some methods that the paper does not compare with. For example, G-Meta is mentioned in the related works but not compared in the experiments. A recent work TENT[2] is not mentioned in related works. As far as I know, the above two approaches can be applied in the problem setting in the paper.
(2) For the approach proposed in the paper, there is no detailed ablation study for the functionalities of each part designed.
(3) It is better to add a case study part to show the strength of the proposed method by an example.

Concerns:

1.The paper consider the node importance among nodes with same label in support set. In 1-shot scenario, how node importance can be used? I also find that the experiment part in the paper does not include the 1-shot paper setting, but related works such as RALE have 1-shot setting, why?

2.The paper says that the theory of node importance can be applied to other domains. I think there should be an example to verify that conclusion.

3.In section 5.3, ‘we get access to abundant nodes belonging to each class’. I do not think this is always true as there might be a class in the training set that only has few samples given the long-tailed distribution of samples in most graph datasets.


[1] Ding et al. Graph Prototypical Networks for Few-shot Learning on Attributed Networks

[2] Wang et al. Task-Adaptive Few-shot Node Classification



**Summary Of The Paper:**

The paper mainly proposes NIML, a few-shot graph node classification method to predict nodes with novel labels. The method considers the node importance among different nodes in a task and neighborhood relationships for a given node. The paper theoretically analyzes the influence of the node importance and verifies the effectiveness of NIML in the experiment part.

**Summary Of The Review:**

[Reject] I reject the paper because the main idea of the paper (node importance) has already been proposed in existing works(GPN) and the experiment for the method is not sufficient enough.

---

> ### Author Response · Authors · 2022-11-17
> **Author Response to Reviewer gYK6 1/2**
>
> We thank reviewer gYK6 for the comments on the paper. We address the concerns and questions as follows.
>
> > **Weakness 1**: I think the work is lack of novelty as the work GPN[1] has already proposed to add node importance score in the calculation of class prototype and the paper only give a theoretical analysis on it.
> >
>
> Please check the summary rebuttal post that addresses common issues, wherein in the first part we discuss the significance of exploring the node importance in graph few-shot problems and the difference between NIML and GPN; and in the second part, we describe the model design motivation and how the model is closely combined with the theoretical results. Thus, our contribution is not only to give a theoretical analysis of it, but also to propose the distance predictor based on the theoretical results.
>
> > **Weakness 2**: The experiment part is not sufficient enough. (1) For few-shot graph node classification problem to predict nodes with novel labels, there are some methods that the paper does not compare with. For example, G-Meta is mentioned in the related works but not compared in the experiments. A recent work TENT[2] is not mentioned in related works. As far as I know, the above two approaches can be applied in the problem setting in the paper. (2) For the approach proposed in the paper, there is no detailed ablation study for the functionalities of each part designed. (3) It is better to add a case study part to show the strength of the proposed method by an example.
> >
>
> (1) We compare the performance with GPN[1] and RALE[2] in the original paper since these two works and NIML all involve the part that distinguishes between nodes inside a single task. We implement experiments on G-Meta[3] and TENT[4] and report the average performance in the table below.
>
> We can see from the results that NIML outperforms G-Meta in each case, but cannot achieve similar results with TENT in all the cases. The main reason is that NIML focuses on the theoretical analysis and method design of node importance in a single task, but TENT explores several different aspects of improving model performance, including task-level adaptation (subgraph for each class), class-level adaptation (parameter adapted GNN to learn prototype), and task-level adaptation (InfoNCE loss). Thus, although we cannot achieve better results over TENT, the improvement from Proto-GCN (the baseline model using mean value to represent prototypes) and GPN (previous work that also explores node importance) to NIML can show the eligibility of NIML on node importance determination inside a single task.
>
> |  | DBLP |  | Reddit |  | Amazon-E |  |
> | --- | --- | --- | --- | --- | --- | --- |
> |  | 5way 3shot | 5way 5shot | 5way 3shot | 5way 5shot | 5way 3shot | 5way 5shot |
> | G-Meta | 72.45 | 76.58 | 64.79 | 65.35 | 64.58 | 69.12 |
> | TENT | 74.71 | 81.34 | 66.67 | 70.42 | 75.21 | 78.91 |
> | NIML | 76.53 | 81.37 | 67.51 | 69.67 | 68.93 | 73.85 |
>
> (2) The ablation study lies in the first part of Section 6.3. In the ablation study. The main part we design in NIML is the importance score predictor, thus we investigate three different importance score calculation methods: score aggregation layer in GPN, direct prediction using GAT, and NIML method. We take the same embedding function for each of the three methods and show that NIML has the best performance among them.
>
> (3) Thanks for the suggestion. We have added a case study part in the appendix to visualize the nodes in a two-dimensional space and color the nodes with their importance score inside a task. The figure shows that nodes with a closer distance to the same-class prototype expectation are assigned higher weights, which is consistent with our theoretical results and method.

---

> > ### Author Response · Authors · 2022-11-17
> > **Author Response to Reviewer gYK6 2/2**
> >
> > > **Concern 1**: The paper consider the node importance among nodes with the same label in the support set. In 1-shot scenario, how node importance can be used? I also find that the experiment part in the paper does not include the 1-shot paper setting, but related works such as RALE have 1-shot setting, why?
> > >
> >
> > Thanks for pointing out the limitation of our method on the 1-shot problem. It is true that NIML targets dealing with $k$-shot problem when $k>1$, since we focus on exploring the important degree of different nodes in support set within a single task. The reason why RALE[2] has a 1-shot setting is that they consider the alignment between tasks by including graph-level absolute location. The method to use NIML on task alignment could be considering task importance using the sum of importance scores of all nodes in each task's support set, thus 1-shot problem can also be dealt with.
> >
> > > **Concern 2**: The paper says that the theory of node importance can be applied to other domains. I think there should be an example to verify that conclusion.
> > >
> >
> > We indeed mention in the conclusion section that the theory can be applied to other domains, not just graphs. The reason is that all the theoretical analysis are performed in the embedding space, which means that no matter graphs or images,  our theory holds as long as they are embedded into a Euclidean space by any embedding function, such as GCN or CNN.
> >
> > > **Concern 3**: In section 5.3, "we get access to abundant nodes belonging to each class". I do not think this is always true as there might be a class in the training set that only has few samples given the long-tailed distribution of samples in most graph datasets.
> > >
> >
> > Thanks for pointing out the lack of samples in some categories brought by the long-tail distribution. In addition to the long-tail distribution, in some datasets, such as Amazon-Electronic and DBLP, the excessive number of labels relative to the number of nodes can also lead to the situation that a portion of labels has relatively fewer data. However, the nodes contained in each category are still relatively sufficient compared to the $k$ nodes in the support set for each few-shot task, thus can provide more correct guidance for computing prototype expectation. We have revised the description in the paper to avoid misleading.
> >
> > We generate the frequency histograms of the labels in the three datasets used in the experiment part and add the histograms to the appendix. Below is a table of descriptions of these three datasets, and we present the number of nodes corresponding to the five lowest frequency categories. We can see that in Amazon-E and DBLP datasets, the lowest frequency is about 100, and it is still relatively larger than 3 or 5 under few-shot settings (N-way 3-shot or N-way 5-shot).
> >
> > | Dataset | # of nodes | # of labels | Five lowest frequency |
> > | --- | --- | --- | --- |
> > | Reddit | 232,965 | 41 | [328, 1003, 1575, 1659, 1696] |
> > | Amazon-Electronic | 42,318 | 167 | [101, 103, 103, 104, 105] |
> > | DBLP | 40,672 | 137 | [101, 102, 103, 105, 106] |
> >
> > **Reference**
> >
> > [1] Ding, Kaize, et al. "Graph prototypical networks for few-shot learning on attributed networks."*Proceedings of the 29th ACM International Conference on Information & Knowledge Management*. 2020.
> >
> > [2] Liu, Zemin, et al. "Relative and absolute location embedding for few-shot node classification on graph."Proceedings of the AAAI conference on artificial intelligence*. Vol. 35. No. 5. 2021.
> >
> > [3] Huang, Kexin, and Marinka Zitnik. "Graph meta learning via local subgraphs."Advances in Neural Information Processing Systems* 33 (2020): 5862-5874.
> >
> > [4] Wang, Song, et al. "Task-adaptive few-shot node classification."Proceedings of the 28th ACM SIGKDD Conference on Knowledge Discovery and Data Mining*. 2022.

---

> ### Comment · Reviewer_gYK6 · 2022-12-08
> **Reply to authors**
>
> I appreciate your response to my questions and concerns. From your response, I can better understand the contribution of the paper.  However, I still feel that the novelty of the paper is not enough as the paper mainly gives the theoretical analysis of a past method and does not add more novel design. And the method has some weaknesses, such as the problem in 1-shot settings and the classes of extreme low frequencies. Also, the performance of the proposed method is not better than TENT, a related work in the same setting. Therefore, I do not think this paper is good enough and I will keep my score.

---

> > ### Author Response · Authors · 2022-12-09
> > **Thanks for you comments**
> >
> > Thank you again for your response. We understand that novelty can be subjective. Other than that, we would like to respectfully make the following clarifications.
> > 1. Our paper **does not** give the theoretical analysis of a past method. In the previous work, GPN takes the assumption that node importance is closely related to the neighbor's importance and uses a score aggregation mechanism to calculate the importance score. However, our theorem quantifies node importance to the distance between a node embedding to its same-class prototype expectation. Our theorem cannot guarantee the nodes with higher learned scores in GPN actually have a closer distance to its same-class prototype expectation. Thus, our theorem conclusion cannot support the previous work GPN.
> > 2. Based on the theoretical analysis, our work **did** add a novel model design. We train a distance predictor in a supervised manner. An attention vector that represents the node community distribution is constructed as the input. And the prototype expectation needed for calculating distance is approximated using the mean value of all same-class node embeddings in the training process.
> > 3. For the weakness of the 1-shot setting, we will compute task importance based on the support set node importance to ensure the 1-shot setting can also benefit from our method. As for the classes of extremely low frequencies, if the classes are in the training set, we can directly omit them to make sure the accuracy of our distance predictor will not be influenced by these extreme cases, but for all the datasets we are using, the extreme cases have not shown up. If the classes are in the test set, we do not need to care about it since we do not need to calculate a "ground truth" distance for supervised learning.
> > 4. For the comparison of a related work TENT, we indeed cannot reach competitive performance with TENT in each case. But our work focuses on the inner-task level node importance, thus it's better to pay attention to the comparison with other node importance related works. Our experiments in the ablation study show the efficiency of our model.

---

### Official Review · Reviewer_QBCi · 2022-11-03

**Confidence:** 3
**Correctness:** 3
**Technical Novelty And Significance:** 3
**Empirical Novelty And Significance:** 2
**Recommendation:** 3

**Clarity, Quality, Novelty And Reproducibility:**

This paper is well polished and easy to understand and follow. It is a novel idea to introduce the theoretical analysis into the importance mechanism.

**Strength And Weaknesses:**

Strengths:
S1, this paper is well presented and easy to understand and follow.
S2, although there are some assumptions, the proposed method is based on a complete theoretical foundation.
S3, the empirical study can demonstrate the effectiveness of NIML.

Weakness:

W1, the theory is based on one assumption that node distance is known. To solve this issue, the authors propose a simple but straightforward solution. It would be better if authors can conduct more experiments to show its eligibility.

W2, as mentioned in Sec. 1, GPN considers the importance of nodes, but it is lack theoretical analysis. The authors are expected to explain why NIML outperforms GPN in Tab. 1 since it also introduces an importance mechanism. (Sec. 6.2 misses such analysis).

W3, authors may need to explain where the results of baselines come from. For example, in the original paper, the acc of GPN in 5-way 5-shot at Reddit, Amazon-Elec, and DBLP are 68.4, 70.9, and 80.1 respectively. But in this paper, these results are 66.6, 70.3, and 78.6. Besides, why another benchmark data Amazon-Clothing is discarded in this paper?

W4, it seems the results of the first part in Sec. 6.3 are copied from Tab. 1. Thus, it is redundant to mention them again. Especially, these baselines use different frameworks. It may be unfair to show the effectiveness of different important mechanisms built on different frameworks. Instead, the authors can implement various mechanisms under one framework, then report the findings of existing important mechanisms.


**Summary Of The Paper:**

To further improve the performance of few-shot graph neural networks (GNNs), the authors investigate the effect of node importance and theoretically demonstrate the effect of node importance on the lower bound of model performance. Then, based on the proposed theory, the authors propose a new method, named Node Importance Meta-Learning (NIML), for the few-shot node classification task. The empirical study shows the effectiveness of NIML.

**Summary Of The Review:**

Although this paper introduces a clear idea, it lacks more discussions on its core idea and some key experimental results.

---

> ### Author Response · Authors · 2022-11-17
> **Author Response to Reviewer QBCi 1/2**
>
> We thank Reviewer QBCi for the insightful comments as well as for acknowledging our theoretical analysis. We address the concerns and questions as follows.
>
> > **Weakness 1**: The theory is based on one assumption that node distance is known. To solve this issue, the authors propose a simple but straightforward solution. It would be better if authors can conduct more experiments to show its eligibility.
> >
>
> Thanks for your suggestion of adding more experiments. From the theoretical conclusion, we know that assigning higher weights to nodes that have a closer distance to the expected prototype embedding will enhance the lower bound of accuracy. Thus we train a **general distance predictor** in a supervised manner in order to tightly connect the theory with the methodology.
>
> To demonstrate the generality and efficiency of NIML, we choose a graph few-shot work: G-Meta[1], and apply NIML to the prototype computation part. G-Meta uses a local subgraph to represent each node and adopts the same framework to different scenarios including a single graph with disjoint labels, multiple graphs with shared labels, and multiple graphs with disjoint labels.
>
> We apply the distance predictor of NIML on the prototype computation part in G-Meta and present the results in three scenarios below. obgn-arxiv is the scenario: single graph with disjoint labels; Tissue-PPI is the scenario: multiple graphs with shared labels; Fold-PPI is the scenario: multiple graphs with disjoint labels.
>
> |  | obgn-arxiv | Tissue-PPI | Fold-PPI |
> | --- | --- | --- | --- |
> | G-Meta | 0.442 | 0.713 | 0.502 |
> | G-Meta+NIML | 0.491 | 0.742 | 0.534 |
>
> The above results clearly show that NIML strengthens G-Meta in all three scenarios, which shows the high efficiency and good adaptability of NIML. The magnitude of the improvement is greatest in the case of a single graph. The reason might be that in the case of a single graph, the distribution of relationships represented by each node's attention vector contains more similar information.
>
> > **Weakness 2**: As mentioned in Sec. 1, GPN considers the importance of nodes, but it is lack theoretical analysis. The authors are expected to explain why NIML outperforms GPN in Tab. 1 since it also introduces an importance mechanism. (Sec. 6.2 misses such analysis).
> >
>
> Thanks for pointing out the lack of analysis on comparing NIML with GPN[2]. Please check the summary rebuttal post that addresses common issues, we include the response to this weakness in the first part of it. We also add the analysis to the revised version.
>
> > **Weakness 3**: Authors may need to explain where the results of baselines come from. For example, in the original paper, the acc of GPN in 5-way 5-shot at Reddit, Amazon-Elec, and DBLP are 68.4, 70.9, and 80.1 respectively. But in this paper, these results are 66.6, 70.3, and 78.6. Besides, why another benchmark data Amazon-Clothing is discarded in this paper?
> >
>
> We implement all the baselines by ourselves and do not directly report the results from original papers to ensure a fair comparison under the same setting. As for the different results of GPN between our paper and the original papers, since the code GPN's authors present on Github is a new version, which uses a two-layer GCN as the score predictor instead of the score aggregation layer proposed in the original paper, we reimplement the method in the original paper and report the results in our paper. Since the standard deviation for each experiment is around 3\% or 4\%, we consider this fluctuation to be reasonable.
>
> We show the results provided by GPN and the results we implement for GPN and NIML below. We can see the fluctuation of GPN in the two versions is within a reasonable range, and NIML outperforms GPN in all the cases.
>
> |  | Reddit |  | Amazon-E |  | DBLP |  |
> | --- | --- | --- | --- | --- | --- | --- |
> |  | 5-way 3-shot | 5-way 5-shot | 5-way 3-shot | 5-way 5-shot | 5-way 3-shot | 5-way 5-shot |
> | GPN (original paper) | 65.5 | 68.4 | 64.6 | 70.9 | 74.5 | 80.1 |
> | GPN (implemented by us) | 65.37 | 66.57 | 65.69 | 70.31 | 74.69 | 78.58 |
> | NIML | 67.51| 69.67 | 68.93 | 73.85 | 76.53 | 81.37 |
>
> We discard the Amazon-Clothing dataset in the paper due to space limitations. We show the results on Amazon-Clothing below. The results show that NIML still outperforms other important baselines in both 5-way 3-shot and 5-way 5-shot few-shot settings.
>
> |  | Amazon-Clothing |  |
> | --- | --- | --- |
> |  | 5-way 3-shot | 5-way 5-shot |
> | Proto-GCN | 73.16 | 76.33 |
> | RALE | 77.42 | 79.58 |
> | GPN | 75.76 | 78.43 |
> | NIML | 77.43 | 82.72 |

---

> > ### Author Response · Authors · 2022-11-17
> > **Author Response to Reviewer QBCi 2/2**
> >
> > > **Weakness 4**: It seems the results of the first part in Sec. 6.3 are copied from Tab. 1. Thus, it is redundant to mention them again. Especially, these baselines use different frameworks. It may be unfair to show the effectiveness of different important mechanisms built on different frameworks. Instead, the authors can implement various mechanisms under one framework, then report the findings of existing important mechanisms.
> > >
> >
> > Thanks for pointing out that the first part in Sec. 6.3 gives the readers a sense of redundancy. We copy part of the results from Table 1 into a separate diagram to highlight the influence brought by different importance mechanisms. In addition, the baselines are **not** under different frameworks, and the embedding function of each of them is the same GCN framework. The only difference among them is as you suggest, the different important mechanisms. We would like to use this part to clearly show the difference between the three different calculation methods of weights, including the score aggregation layer in GPN, directly using GAT, and NIML. The results show that NIML outperforms other importance calculation methods.
> >
> > **Reference**
> >
> > [1] Huang, Kexin, and Marinka Zitnik. "Graph meta learning via local subgraphs."Advances in Neural Information Processing Systems* 33 (2020): 5862-5874.
> >
> > [2] Ding, Kaize, et al. "Graph prototypical networks for few-shot learning on attributed networks."Proceedings of the 29th ACM International Conference on Information & Knowledge Management*. 2020.

---

### Author Response · Authors · 2022-11-17
**Summary to All Reviewers and AC 1/2**

We thank all reviewers for your constructive comments and insightful feedback. We reply to some common concerns below to better explain our work.

> Why do we choose to investigate the methods of determining node importance even if a previous work GPN has explored this direction? What is the difference between NIML and GPN?
>
1. Although GPN[1] has explored a score aggregation method to include node importance scores in the prototype computations, we believe that, for few-shot graph problems, **the determination of node importance in the support set still has a lot of issues to study**. In general few-shot problems, since the support set for each task has very few data points, distinguishing data points of different importance degree when computing prototypes can reduce the negative impact of sampling randomness. In particular, in graph data, the properties of non-Euclidean data, such as node connectivity and graph structure, make more information available for exploring node importance, thus it is worth investigating how to utilize these properties to assist in learning the node importance.
2. Even though, both NIML and GPN make an effort to compute weighted prototypes, the two methods are designed with **different intentions**.
**NIML** starts with a theoretical analysis, quantifying the node importance as the distance from the node to its same-class prototype expectation and concludes that assigning higher weights to nodes with closer distance will enhance the lower bound of model accuracy. After that, NIML adopts the idea that the distribution of the relationship between a given node and its neighbors can reflect the node importance and then construct an attention vector that depicts the relationship distribution as input to predict the distance in a supervised manner, further learning the node importance.
While **GPN** adopts a different view that assumes the importance of a node is highly correlated with its neighbor's importance and derive a score aggregation mechanism using GAT as the backbone, which has similar characteristic to message passing that relies on graph homophily. We think this is the main reason why NIML outperforms GPN as shown in Table 1.

---

> ### Author Response · Authors · 2022-11-17
> **Summary to All Reviewers and AC 2/2**
>
> > What is the main contribution of NIML? What is the motivation for model design? How does the model design closely related to the theoretical conclusion?
> >
>
> The main contribution of NIML is not only the theoretical analysis but also the importance score predictor which is closely integrated with theoretical conclusions.
>
> In the theoretical part, we quantify the node importance as the distance from a given node to its same-class prototype expectation and conclude that in the computation of each prototype, compared to computing the mean of node embeddings in the support set, assigning higher weights to the nodes with closer distance to its same-class prototype will enhance the lower bound of model accuracy.
>
> Now that closer distance means higher importance, **the theoretical results prompt us to explore how to learn the distance** between a given node and its same-class prototype expectation. The calculation of distance $d(f_{\phi}(x_v),\mu_c)$ involves two elements: the embedding $f_{\phi}(x_v)$ of given node $v$ and the same-class $c$'s prototype embedding expectation $\mu_c$. The embedding $f_{\phi}(x_v)$ can be calculated by the embedding function like GCN, but $\mu_c$ is unknown in the test phase. Since there is no way to calculate $\mu_c$, is it possible to predict the distance directly?
>
> Then it occurred to us that although $\mu_c$ can not be computed during testing, it can be computed during training. Since the definition of $\mu_c$ is the expectation of class $c$'s prototype embedding, it can be approximated using the mean value of relatively abundant embedding of nodes that belongs to class $c$. As we know, during the training phase, we have access to all nodes in the training set, which means we have access to all nodes belonging to class $c$, and $\mu_c$ can then be estimated. Up to now, for each node involved in the training process, we can calculate both its embedding and the prototype expectation of the class it belongs to, and we naturally can obtain the distance between the node and its same-class prototype expectation.
>
> Next, we consider that we can design a supervised distance prediction model: given some information about a node, output the distance between the node and the same-class prototype expectation. The distance we calculated using the method mentioned in the previous paragraph can be taken as the ground truth. Knowing the output of the model, what kind of node information can be used as input to the model? We thus resort to the idea that the **distribution of relationships** between a given node and its neighbor can reflect the importance of a node, and construct an attention vector as the model input to represent the relationship distribution.
>
> After training the distance predictor, in the test phase, we can predict an importance score for each node in a task given the attention vector. The closer the distance is, the more important the node is, and the higher weight will be assigned to that node when computing the prototype for each category.
>
>
> **Reference**
>
> [1] Ding, Kaize, et al. "Graph prototypical networks for few-shot learning on attributed networks."Proceedings of the 29th ACM International Conference on Information & Knowledge Management*. 2020.

---

### Decision · Program_Chairs · 2023-01-20

**Decision:**

Reject

**Justification For Why Not Higher Score:**

The motivation and theoretical results are not well explained and motivated.

**Justification For Why Not Lower Score:**

N/A

**Metareview: Summary, Strengths And Weaknesses:**

Adding weights to learning samples have been used in many machine learning tasks, and this paper extends the same idea to the scenario of few-shot node classification. The proposed method starts from a theoretical analysis which shows assigning higher weight to the data point that has closer distance to the prototype expectation will increase the lower bound of accuracy. Then, the authors propose a framework for node classification by introducing an importance score predictor. Experiments results show the proposed method is promising.

Strength
- This paper is well presented and easy to understand and follow.
- Empirical results are promising

Weakness
- Methodology design is not very novel, given existing works already consider sample weights (especially in the domain of few-shot graph learning).
- The usage of the theoretical results is not clear. There are concerns on how distance is estimated, how the lower bound indicate the practical performance. Besides, as authors also explained, the results are not specified to graph learning, it is suggested to have graph properties in the results or adding extra experiments outside graph domain.